# Long Noncoding RNA *HCP5*, a Hybrid HLA Class I Endogenous Retroviral Gene: Structure, Expression, and Disease Associations

**DOI:** 10.3390/cells8050480

**Published:** 2019-05-20

**Authors:** Jerzy K. Kulski

**Affiliations:** 1Faculty of Health and Medical Sciences, UWA Medical School, The University of Western Australia, Crawley, WA 6009, Australia; kulski@me.com or yurek.kulski@uwa.edu.au; Tel.: +61-047-799-9943; 2Department of Molecular Life Science, Division of Basic Medical Science and Molecular Medicine, Tokai University School of Medicine, Isehara 259-1193, Japan

**Keywords:** HCP5, lncRNA, MHC, HLA, human endogenous retrovirus (HERV), cancer, autoimmune diseases, competing endogenous RNA (ceRNA), human immunodeficiency virus (HIV), human papillomavirus (HPV)

## Abstract

The *HCP5* RNA gene (NCBI ID: 10866) is located centromeric of the *HLA-B* gene and between the *MICA* and *MICB* genes within the major histocompatibility complex (MHC) class I region. It is a human species-specific gene that codes for a long noncoding RNA (lncRNA), composed mostly of an ancient ancestral endogenous antisense 3′ long terminal repeat (LTR, and part of the internal *pol* antisense sequence of endogenous retrovirus (ERV) type 16 linked to a human leukocyte antigen (HLA) class I promoter and leader sequence at the 5′-end. Since its discovery in 1993, many disease association and gene expression studies have shown that *HCP5* is a regulatory lncRNA involved in adaptive and innate immune responses and associated with the promotion of some autoimmune diseases and cancers. The gene sequence acts as a genomic anchor point for binding transcription factors, enhancers, and chromatin remodeling enzymes in the regulation of transcription and chromatin folding. The *HCP5* antisense retroviral transcript also interacts with regulatory microRNA and immune and cellular checkpoints in cancers suggesting its potential as a drug target for novel antitumor therapeutics.

## 1. Introduction

The human major histocompatibility complex (MHC), also known as the human leukocyte antigen (HLA), covers 0.13% of the human genome and spans ~4 Mbp on the short arm of chromosome six at position 6p21 within a region that contains more than 250 annotated genes and pseudogenes [1,2]. The classical class I and class II regions within the MHC have extensive patterns of linkage disequilibrium (LD), and a high degree of single nucleotide polymorphisms (SNPs) at the HLA genes can differentiate worldwide populations [1,3,4,5]. HLA polymorphisms are a crucial determinant of the adaptive immune response to infectious agents, allograft success, or rejection and self/nonself immune recognition that can contribute to more autoimmune diseases than any other region of the genome [1,2,6,7,8]. Apart from the adaptive immune response, MHC class I molecules have a role in brain development, synaptic plasticity, axonal regeneration, and immune-mediated neurodegeneration [9,10,11,12]. At least half of the molecules encoded by this highly polymorphic locus are involved in antigen processing and presentation, inflammation regulation, the complement system, and the innate and adaptive immune responses, highlighting the importance of the MHC in immune-mediated autoimmune and infectious diseases [1,2]. Polymorphisms expressed by the MHC genomic region influence many critical biological traits and individuals’ susceptibility to the development of chronic autoimmune diseases such as type I diabetes, rheumatoid arthritis, celiac disease, psoriasis, ankylosing spondylitis, multiple sclerosis, Graves’ disease, schizophrenia, bipolar disorder, inflammatory bowel disease, and dermatomyositis [2,6,7,8]. Furthermore, different viral infections and cancers are associated strongly with the suppression of MHC genomic expression activity, particularly in the region of the MHC class I and class II loci [6,13,14,15].

There are tens of thousands of genomic loci that express microRNA (miRNA) [16] and lncRNA [17,18,19,20], but only about 50 have been investigated in any great detail with respect to their role in the regulation of the immune system and disease [21,22,23,24]. Although there are many miRNA and lncRNA loci within the MHC genomic region, they have been ignored largely in favor of studies on polymorphisms of the HLA class I and class II gene loci in health, disease, and transplantation cell/tissue/organ typing [25,26]. This review focuses on the structure and function of only one of these HLA lncRNA, the *HCP5* lncRNA, which is located between the *MICA* and *MICB* genes and ~105 kb centromeric of the *HLA-B* gene.

In 1993, Vernet et al. [27] discovered a novel coding sequence belonging to a new multicopy pseudogene family P5 that they mapped within the HLA class I region and named *P5-1* (alias for *HCP5*). They found that it expressed a 2.5-kb transcript in human B-cells, phytohemagglutinin-activated lymphocytes, a natural killer-like cell line, normal spleen, hepatocellular carcinoma, neuroblastoma, and other non-lymphoid tissue, but not in T-cells. *HCP5 (P5-1*) appeared to be a hybrid sequence created by nonhomologous recombination between two pseudogenes or nonmobile genetic elements that possibly produced a protein comprising 219 amino acids (aa’s) [28]. A few years later, the *HCP5 (P5-1)* gene was mapped precisely to a region between the *MICA* and *MICB* genes and downstream, at the centromeric end, of the two classical HLA class I genes *HLA-B* and *HLA-C* (Figure 1) [29]. In 1999, Kulski and Dawkins [30] used the computer programs Censor and RepeatMasker and dot-plot DNA and RNA sequence analyses to demonstrate that the *HCP5* gene sequence and its transcripts were composed mainly of the 3′LTR and *pol* sequences of an ancient *HERV16* insertion, which was a member of the HERVL or class III category of endogenous retroviruses (ERVs) in the human and mammalian genomes [31,32].

Because *HCP5* expressed an antisense transcript that was complementary to retrovirus *pol* mRNA sequences and a 3′LTR, Kulski and Dawkins [30] suggested that it might have a role in immunity to retrovirus infection. They considered that the lncRNA of *HCP5* might hybridize with retroviral sense mRNA sequences to suppress viral transcription, translation, and transport. Eight years later, a single-nucleotide polymorphism (rs2395029) in the *HCP5* gene was associated with *HLA-B*57:01* and correlated with a lower HIV-1 viral set point [33], indicating that these two alleles within a particular haplotype may have a role in viral control [34]. However, when Yoon et al. [35] tested the antisense/antiviral hypothesis for *HCP5* by infecting TZM-bl cells in vitro with HIV-1 and plasmids expressing high levels of *HCP5* transcripts, they observed no restriction with infectivity throughout the viral life cycle. They concluded from their findings that the *HCP5* gene had no direct antiviral effect, and that the association of an *HCP5* variant with viral control most likely was due to an *HLA-B*57:01*-related effect or other functional variants in the haplotype or both. In fact, it appears that the role of *HCP5* in immunity and human disease is far more complex than previously envisioned, and that its antiviral affects might occur by way of some secondary mechanisms such as the possible involvement of miRNA inhibition rather than by hybridization of the *HCP5* transcript with the complementary viral *pol* transcripts.

During the last two decades, *HCP5* SNPs have been associated with many different diseases in genome-wide association studies (GWASs), gene expression studies, and cancer studies investigating tissue and cellular biomarkers of tumor progression and inhibition. To better understand the genetics, molecular biology, and functions of *HCP5*, this paper reviewed the available data and literature on the genomic organization, structure, and function of the *HCP5* gene (HLA complex P5 (non-protein coding), HGNC:21659) in health and disease (MIM:604676), particularly its association with autoimmune diseases, cancer, and infections by way of its endogenous interactions with miRNA and various gene targets. Appendix A lists the online databases and repositories that were searched and interrogated to find the available data on *HCP5*. Appendix A lists a summary of the downloaded functional associations in all of the datasets linked with Harmonizome, which is an integrated knowledge base connecting big data with a collection of information about genes and proteins from 114 datasets provided by 66 online resources (Appendix A).

## 2. *HCP5* Genomic Organization, Gene Structure, and the *HERV16* Antisense Transcript

### 2.1. Human MHC Class I Genomic Structure as Duplication Blocks and HCP5 Location within an HERV16 Duplicated Sequence

The HLA class I genomic region within chromosome six is composed of highly polymorphic frozen blocks [36] that were generated by a series of duplication, deletion, and various genomic rearrangement events during mammalian and primate evolution. Numerous crossover events of a tripartite gene combination (duplicon) of *MIC*, *HERV16*, and *HLA class I* generated the duplication blocks and the HLA class I structural organization of the HLA class I and MIC genes within the beta and alpha blocks of the MHC genomic region in an unidentified ancestor [37]. The ancient endogenous retroviral sequence *HERV16* is repeated at least twelve times, along with the HLA class I coding and noncoding sequences within the alpha and beta block of the MHC class I region, and it appears, along with other retroelements, to have been a recombination site for many of the duplication events involving unequal crossovers [37,38]. The beta-block of approximately 362 kb between the *POU5F1* and *MCCD1* genes harbors the *MICA, MICB, HLA-B*, and the *HLA-C* genes as well as the *HCP5* gene that is centromeric of a short, fragmented *HLA-X* pseudogene (645 bp) and telomeric of the *HCG26* long noncoding RNA gene (Figure 1). The *HCP5* gene is located between the *MICA* and *MICB* genes at the centromeric end of the HLA beta block, and it also has a neighboring 3538 bp *HERV16* fragmented duplication (alias P*5.8*) at chr6:31383746-31387283 between the *HLA-S* (alias H*LA-17*) pseudogene (919 bp) and the *MICA-AS1* RNA gene [30,38]. The *HCP5* gene is different from all the other duplicated *HERV16* sequences in the MHC in that the ancestral endogenous antisense 3′LTR and part of the internal *pol* antisense sequence of *HERV16* (exon 2) is linked to an HLA class I promoter and leader sequence at the 5′-end (exon 1) that regulates its expression (see Section 2.5).

There are at least four *HCP5* gene sequence variants in the genomic databases to consider: *HCP5-1* with 2630 bp (or 2432 to 2658 bp); *HCP5-2* of 23 kb; *HCP5-3* of 575 bp; and *HCP5-4* of 465 bp ((HGNC: 21659, Entrez Gene: 10866, Ensembl: ENSG00000206337, OMIM: 604676, UniProtKB: Q6MZN7, Vega: OTTHUMG00000031282, and GCID: GC06P031400). The original *P5.1* sequence (GenBank L06175) reported by Vernet et al. [27] was 2535 bp. According to the online Ensembl database (February 2019), *HCP5* has four transcript or splice variants and one gene allele that is associated with 64 phenotypes. In this review, the *HCP5* gene of 2,630 bp is defined by the NCBI Gene ID: 10,866 (updated 31 January 2019) and by its location on the GRCh38.p12 assembly (annotation release 109 in March 2018) at chr6:31,463,180–31,465,809 (NC_000006.12) with a transcript length of 2547 bp, including eight adenines of the RNA polyA tail (NCBI ncRNA reference sequence NR_040662.1). It spans 2630 bp of a 5153 bp (nt positions from 717 to 3346) extended *HCP5* gene reference with 5′ and 3′ noncoding regions (GenBank AB088109.1) and the chromosome six reference sequence NC_000006.12. DNA sequence alignment of NR_040662.1 and AB088109.1 or NC_000006.12 revealed that the *HCP5* gene is composed of two exons: a 100 bp exon 1 and a 91 bp intron and 2355 bp exon 2. The HERV16 sequence begins at nucleotide position 150 within exon 2 of *HCP5* and occupies most of exon 2 of the HCP5 RNA sequence NR_040662.1.

Instead of simply annotating the *HCP5* gene at positions 31,463,180–31,465,809 on chromosome six, the Ensembl human genome assembly GRCh38.p10 that is linked to various other gene and genomic databases (OMIM, UniProtKB, HGNC, Vega, and GCID) complicated the genomic location by placing the variant (Ensembl: ENSG00000206337) at the extended genomic coordinate of 6:31,400,702 to 31,477,506 (76,805 nt). This extended genomic coordinate for the *HCP5* large variant is confusing because the 5′-end is within the *MICA* gene sequence ~76,000 bp from the transcription start site and overlaps *MICA* by 60% rather than within the 5′-UTR promoter region closer to the transcription start site of the 2630 bp *HCP5* variant (Figure 1, Table 1). Furthermore, these extended genomic coordinates for the large *HCP5* variant can be misleading with respect to the location of SNPs within the *HCP5* gene and the transcription factor regulatory elements within the 5′-UTR of the *HCP5* gene. For example, the *HCP5* SNP rs6938467 (T/G) that is associated significantly with Kawasaki disease in Korean children [39] is positioned at chr6:31,440,139 (GRCh38.p12) well outside the *HCP5* RNA reference sequence (NR_040662.1) and upstream of the *LINC01149* locus starting within the *MICA* gene in the genomic location of chr6:31,400,702–31,465,809 and ENSG00000206337. Therefore, this expanded version of the large *HCP5* variant sequence (ENSG00000206337) with a genomic length of 65,108 to 76,000 bp has been excluded from this review except for occasionally noting whether or not the published *HCP5* SNPs in disease association studies lie inside or outside the 2630 bp version of the *HCP5* gene sequence.

### 2.2. HCP5 RNA, a Complementary Sequence of ERV16 (an ERV Class III (ERVL))

The gene structure of *HCP5* is unusual because it is a hybrid sequence of an *HLA* class I gene fragment in exon 1 and the fragmented portions of the 3′LTR and internal sequence of *ERV16* in exon 2 [30], annotated by RepeatMasker as *3′LTR16B* and complementary (minus strand) *ERV3-16A3* (Table 1). The entire *ERV16* sequence from *5′LTR16B* to *3′LTR16B* with the internal *ERV3-16A3* is located on the reverse strand between chr6:31,469,592 to 31,463,420 (6173 bp), and it is 2.4 times longer than the *HCP5* gene sequence of 2630 bp (Figure 1). The internal *ERV3-16A3* sequence contained between the flanking LTRs has about 60% similarity to *HERVL*, mainly within the *pol* gene region and with only a slight identity to parts of the *gag* gene. An assumed primer binding sequence (5′ -TGGCTTCAGGAGTGGTCC-3′) for leucine-tRNA is present two nucleotides downstream from the 3′ end of the *5′-LTR16B* sequence, which has identity with 15 of 18 nucleotides of *HERVL* [30]. Thus, the *HCP5* 2547 bp transcript of the reference sequence NR_040662.1 terminates about a third of the way from the *3′LTR16B* to the *5′LTR16B2* of the *ERV16* sequence, with the two *LTR16B* sequences possibly acting as promoters and/or enhancers for *HCP5* and various other expressed sequences within the vicinity [30]. The *ERV16* sequence downstream of *HCP5* is also interrupted with insertions from other retroelements including *AluSp*, *THE1B*, and *AluSx* (Table 1), indicating that this *ERV16* sequence is at least as old as the wave of *AluSp/x* insertions that occurred in primates about 37 million years ago [40].

### 2.3. HCP5 and the Human Papillomavirus (HPV) Minor Structural Protein Interacting Protein (PMSP) Gene

A surprising finding during a BLAST search of *HCP5* sequences in GenBank was that a portion of the *HCP5* RNA (NR_040662.1, 1844–2011 bp) shares 99% identity with the 186 bp sequence of the papillomavirus minor structural protein interacting protein *(PMSP)* gene sequence (AJ437509.2) (Figure 2) that is only 7% of the length of the entire *HCP5* sequence. *PMSP* codes for a 61 aa peptide (UniProtKB-Q8TCT4) that interacts with the capsid L2 protein of HPV1, HPV11, and HPV16 in the cytoplasm, and both proteins are transported into cellular nuclear dots for viral assembly [41]. In this regard, the 7% of *HCP5* appears to be the solitary genomic locus for *PMSP* gene expression. Alternatively, the *PMSP* gene might be found at a different genomic locus that is still to be identified. The start of the *PMSP* gene (1 to 22 bp) has 100% identity with *HCP5* (NR_040662.1) at nt positions 37 to 58, and the remainder of the *PMSP* gene has 98% identity with 164 bp of *HCP5* (NR_040662.1) at nt positions 1847 to 2012 located in the *ERV16* portion of *HCP5* (Figure 2). Thus, both the *PMSP* gene and the *HCP5* gene of the human genome share the same part of the *ERV16* sequence, possibly generated as a hybrid at some time before or early in primate history. The 164 bp portion of the *PMSP* sequence within *HCP5* is also present in macaque monkeys with 91% identity, but not in the chimpanzee or gorilla where the *HCP5* sequence between the two *MIC* genes has been deleted in available sequences [42]. The *PMSP* gene also has about 88% identity for 141 bp within the multicopy *Homo sapiens* zinc ribbon domain containing one antisense pseudogene (*ZNRD1ASP*) transcript variant 1, noncoding RNA (2450 bp) at three locations within the class I region of the MHC on chr6:31,465,114–31,465,281 in *MICB*; 30,001,688–30,001,549 between *HLA-J* and *ZNRD1*; and 29,713,247–29,713,129 between *HLA-F* and the *IFITM4P* noncoding RNA gene. The *PMSP* portion of the *HCP5* transcript might be upregulated during warts infections in humans; plantar warts are associated significantly with the SNP rs9267257 that is located at 6:31488485 within the *LOC102725068*, an intergenetic variant position between *HCP5* and *MICB* [43]. Conversely, Karim et al. [44] showed that episomal copies of *HPV16* and *HPV18* in undifferentiated keratinocytes downregulated the expression of *HCP5* and the *HLA-A, -B, -C*, and -*G* gene products that were involved with the antigen presenting pathway in order to allow persistence of latent infections. However, expression of the *HCP5* gene that was downregulated substantially by human papillomavirus (HPV) infection was again upregulated 2.2-fold in HPV-infected keratinocytes after 24 h of treatment with polyI:C compared to greater than three-fold upregulation in HPV-uninfected cells during the same period [44]. Further studies are needed to confirm the possible association between *HCP5*, *PMSP*, and the assembly of the HPV capsids in various papillomaviral infections.

### 2.4. HLA Class I Leader Sequence and Cytomegalovirus (CMV) Signal Peptides

The *HCP5* RNA sequence has an open reading frame (ORF) that may code for a peptide of 132 aa (Q6M2N7.1) with a domain resembling HLA class I signal peptides [28,30]. It has another ORF with the first 24 of 52 aa of the putative HCP5 peptide starting from methionine and sharing identity with at least 18 of 24 aa of the leader peptide of HLA-A, -B, and -C, whereas the remaining 28 aa have no sequence similarity to any known protein [30]. The first 23 aa, RVMESRTLLLLFSGAVALIOTWA, of the putative 52 aa HCP5 peptide sequence shares at least 65% (15/23 aa) identity and up to 78.3% (18/23 aa) identity with the leader peptides of HLA-A, -B, and -C. However, it is only the MESRTLL portion of the sequence that is found at the start of the 61 aa peptide that is encoded by the *PMSP* gene (Figure 2). Also, the HCP5 peptide sequence VMESRTLLL shares 74% identity with the VMAPRTLIL sequence within the signal peptides of HLA-A and the UL40 protein of the human cytomegalovirus (HCMV), which binds and stabilizes cell surface expression of both the endogenous HLA-E and the HCMV-encoded MHC-I homologue gpUL18 to regulate differentially two distinct natural killer cell evasion pathways [45,46]. HLA-E and gpUL18 bind a restricted subset of peptides derived from the leader peptides of other class I molecules, but whether they bind the putative HCP5 leader peptide or the PMSP leader peptide is not known.

### 2.5. HLA-Class I Gene Promoters and Enhancers within HCP5

The *HCP5* genomic sequence has a putative HLA-class I gene promoter for transcriptional activation of the *P5-1* mRNA (Figure 1 and Appendix A) that is linked to a 159 bp sequence homologous to exon 1 (signal peptide) and part of intron 1 of classical class I genes [28,30]. Appendix A shows the Clustal W alignment of 360 bp of sequence, including the promoter regions of the *HCP5* and *HLA-A* genes, approximately 120 bp of the start of the *HCP5* RNA sequence, and 159 bp of the start of the *HLA-A* mRNA sequence. The 5′ nonretroviral promoter region of the *HCP5* gene has at least 88% identity with the promoter regions of various class I HLA genes including *HLA-A, -B, -C, -E, -F*, and -*G* [44] (Appendix A). There is also 84% to 87% identity between the *HCP5* sequence and the first ~159 bp of the HLA class I transcripts.

The *HCP5* gene promoter and the associated exon and intron 1 of MHC class I genes are located within a 1075 bp genomic region between the *MER4C* and *HERV* sequence (Table 1). The putative gene promoter and the peptide-coding region are linked to the *3′LTR16B* fragment and are in reverse orientation to the *HLA-X* pseudogene (Figure 1) that is positioned between *MER4C* and *MER21B* fragment within the *HCP5* genomic sequence [30]. The HLA-class I gene promoter of *HCP5* has an intact TATA box, CAAT box, and regulatory response sequences for interferon, kappa B1, and kappa B2 that have been described for the regulatory complex region of class I promoters [28,30,47]. Numerous copies of a motif (AGAAA) for heat shock transcription factor are distributed in forward and reverse orientation within the *HCP5* gene promoter region, and a potential cap signal sequence (5′-GACTCAGATTC-3′) for transcription initiation is located 26 nt downstream from the TATA box [30].

Although few experiments have been performed directly on the *HCP5* promoter, the high sequence similarity with the HLA class I promoters suggests that it is regulated and expressed similarly to *HLA-A, -B, -C, -G*, and -*F* but with differences dependent on polymorphisms and mutations providing sequence variants [47] (Appendix A) and tissue-specific effects on expression [48]. Regulatory elements and transcription factors associated with the MHC class I promoter region were reviewed recently [47,48,49,50] and highlighted that the MHC class I promoter region with the MHC class I NF-**κ**B enhanceosome, together with the RFX-complex, ATF1/CREB, and the NFY-complex (binding the SXY-module), activated MHC class I transcription. Additionally, type II interferons (IFN-**γ**) can activate MHC class I transcription by upregulation of *IRF1*, which bind to the interferon-sensitive response elements (ISREs) of the MHC class I promoter [51,52]. Gene Expression Omnibus (GEO) expression profiles reveal that *HCP5* gene expression is regulated by IFN-γ (Table 2) along with other MHC class I genes [53,54,55]. Recently, TGFbeta was shown to induce over-expression of the *HCP5* gene and activity of the S-mothers against decapentaplegic homolog 3 (SMAD3) protein complex in adenocarcinomas [56]. Also, the ubiquitous transcription factor SP1, which is known to regulate MHC class I gene expression both constitutively and in a tissue-specific manner [48], was shown to upregulate HCP5 expression and induce the development of osteosarcoma [57]. Moreover, the osteosarcoma study also found that the transcription factors STAT3, E2F1, and NF-**κ**B1 had no effect on *HCP5* expression. In contrast, NF-**κ**B binding to Enhancer A is necessary for both constitutive and induced MHC class I expression [58], and NF-**κ**B-induced MHC class I expression is most prominent for the *HLA-A* locus, which contains two NF-**κ**B binding sites in its Enhancer A region [49,59]. Figure 3 shows the comparison and strong correlation between *HCP5* and *HLA-B* expression in the same 27 tissues that were the results of the NCBI BioProject PREJEB4337. In general, *HLA-B* transcription is expressed at much higher levels with a range of 1 to 590 reads per kilobase of transcript per million mapped reads (RPKM ) than *HCP5* transcription with a range of 1 to 39 RPKM.

The class I transactivator (CITA) NLRC5 (alias NOD4, NOD27, and CLR16.1) induces genes involved in the MHC class I-dependent antigen processing and presentation pathway; it activates the promoters of MHC class I and related genes by co-binding within the SXY module of MHC class I to form a CITA enhanceosome and induces the expression of classical MHC class I, nonclassical class I, the beta-2 microglobulin, immunoproteasome component (*PSMB9*), peptide transporter (*TAP1*) [61,62,63], and *HCP5* [64].

The *HCP5* RNA also contains the *3′LTR16B2* and a substantial portion of the internal *ERV16* sequence (Table 1). It should be noted however that there are ~860 *ERV16* copies of *LTR16B2* with the *ERV16* internal regions and ~18,100 copies of the solitary *LTR16B* within the human genome [31]. These ERVs provide an enormous reservoir of autonomous gene regulatory modules, some of which play important roles in normal regulation of genes and gene networks and influence the transcription of host genes. At least one of these *5′LTR16B* sequences is known to act as a promoter for the activation of the isoform anaplastic lymphoma kinase (ALK) of the receptor tyrosine kinase (RTK) in ~11% of skin cutaneous melanomas [65]. The regulatory elements of HERV/LTRs tend to locate near to the genes involved in immune responses in order to interact with them, indicating that these regulatory elements play an important role in controlling the immune regulatory network [66]. Therefore, the possibility exists that the *3′LTR16B2* antisense sequence within the lncRNA of *HCP5* might interact with one or other *LTR16B2* sense sequences distributed across the human genome to regulate the expression of *LTR16B2*-regulated genes outside the MHC genomic region.

### 2.6. HCP5-CCCTC-Binding Factor (CTCF) Connection and Chromatin Structural Regulation

There are two high-probability CCCTC-binding factor (CTCF) binding sites within the body of the *HCP5* genomic sequence that demarcate the likely position of open and closed chromatin (Figure 4). The *CTCF* gene (ID: 10664) encodes the transcriptional repressor CTCF, also known as 11-zinc protein or CCCTC-binding factor, which is involved in many cellular processes including transcriptional regulation, insulator activity, regulation of interferon-gamma (IFNg) induction of MHC class I and class II gene expression [49,66,67], V(D)J recombination in B and T cells [68], and regulation of chromatin architecture [69]. The primary role of CTCF is in the regulation of the 3D structure of chromatin by binding together strands of DNA to form chromatin loops and anchor DNA to the nuclear lamina [70], but it also has an essential role in blocking the interaction between enhancers and promoters to prevent oncogenesis [71]. This nuclear protein is able to use different combinations of the zinc finger domains to bind different DNA target sequences and proteins, and depending upon the context of the site, the protein can interact with a histone acetyltransferase (HAT)-containing complex as a transcriptional activator or a histone deacetylase (HDAC)-containing complex as a transcriptional repressor. If the protein is bound to a transcriptional insulator element, it can block communication between enhancers and upstream promoters, thereby regulating imprinted expression.

The insulator factor CTCF controls MHC class I and II gene expression and is required for the formation of long-distance chromatin interactions [49,66,67]. A search of the ENCODE database revealed that the proximal end of the *HCP5* sequence had two CTCF binding sites in addition to peak clusters of DNase I hypersensitivity, H3K4me3, and H3K27ac with Z scores >1.8 (Figure 4). In this regard, the first 100 nucleotides of *HCP5* exon 1 has a DNase I hypersensitivity region of CpG island DNA (Figure 1f, Table 1) that has 100% sequence homology with CpG island DNA on chr 22 (NCBI, AJ236677.1) [72]. A number of published studies have shown that there are differential methylation sites such as CTCF within the *HCP5* gene, and that hypomethylation is associated strongly with the regulation of a number of phenotypes and diseases (see Section 4, *HCP5* methylomics). The two experimentally unverified CTCF binding sites in the Ensembl database were at the *HCP5* core positions ENSR0000078786 and ENSR00000195544. Of the 123 cells studied for position ENSR0000078786, 2 were active, 28 inactive, 3 poised, 6 repressed, and the regulatory activity was unknown for 84 cell types. The cells in the active or poised positions for ENSR0000078786 were DND-41 (a human leukemic T-cell line with a p53 mutation) and MCF7, and those for ENSR00000787587 were A549 (an epithelial lung carcinoma), a keratinocyte, and MCF7. The presence of CTCF binding sites at the *HCP5* locus (Figure 4) is noteworthy given the importance of the CTCF transcription factor in diverse genomic regulatory functions including regulating chromatin structure, gene expression, and development of various acute and chronic diseases. IFNg probably activates the CTCF binding sites at the *HCP5* locus [51,52,53,54,55], but this requires experimental confirmation.

### 2.7. Structural Deletion of HCP5 and MICA Genes

Examination of the *HCP5* locus in the Database of Genomic Variants (Appendix A) revealed that this gene region is the site of copy number variants (CNVs) with multiple reported deletions (losses) and duplications (gains). The deletion of this locus is associated mostly with a null haplotype for the *MICA* and *MICB* genes and LD with *HLA-B48:01*—reported initially to occur in 3.7% of East Asian individuals [42,73]. Subsequently, the *HLA-B48:01* allele was genotyped at a frequency of 26% or higher in some ethnic groups such as the Ami of Taiwan [74] or the Angaite Amerindian community in Paraguay [75]. In a cohort of 2026 disease-free Caucasians and African Americans, 38 deletions and four duplications of the *HCP5* locus were detected [76]—29 of the deletions had the same apparent breakpoints and seven had different breakpoints. For one of the reported sample sets of 65% Caucasians and 34% African Americans, deletions were detected in 31 African Americans and only seven Caucasians [76]. In another study of 1854 samples that represented 790 Europeans and 1064 individuals from 51 different populations, *HCP5* locus deletions were detected in 34 ethnically diverse individuals (e.g., Chinese, Japanese, Palestinian, Pakistani, and Yoruban) and in only three Europeans [77]. The unforeseen deletion of *HCP5* might interfere with the *HCP5* genotyping assay for abacavir hypersensitivity [78]. However, an alternative and simple proxy assay for *HLA-B*57* and *HLA-B*48:01* could be the structurally polymorphic *AluMICB*, which is a lineage indel within intron 1 of the *MICB* gene [79].

The *HCP5* genomic region has been deleted in gorillas and chimpanzees because of recombination and fusion between the *MICB* and *MICA* genes [80,81,82]. On the other hand, the transcriptional activity and function of the *P5.1.1/HCP5*-like ortholog in the Macaque monkey is not known, although its structure has diverged substantially from that of *HCP5* in the human. The human lineages studied to date carry two functional MIC genes, *MICA* and *MICB*, and a number of unprocessed pseudogenes (*MICC*, *D, E, F*, and *G*) [1,2]. *MICA* and *MICB* appear to have diverged from each other 33–44 million years ago [83,84], before the divergence of chimpanzee and human. Nevertheless, deletion of *HCP5* and some forms of the *MIC* genes from humans, gorillas, and chimpanzees highlights the plasticity of this intergenic genomic region in different primate species.

In Japan, the *HLA-B*48:01* haplotype was estimated to have a 3.2% frequency with 62% of these haplotypes carrying the *MIC/HCP5* null configuration. Based on the current size of the Japanese population and a homozygote rate of 0.1025%, more than 80,000 individuals could be expected to have the homozygous *HLA-B*48:01* haplotype [85]. Apparently, the *MIC/HCP5*-deficient individuals have no overt clinical symptoms of endangered health, unlike those without MHC class I or class II molecules who suffer from mild to severe immune deficiency syndromes and diseases. In an unpublished study, homozygous *HLA-B*48:01* individuals were reported to have no immunological defect in T cell or B cell populations, no change in T cell or B cell receptor repertoire, no obvious change in immunological signals, and no known susceptibility to any diseases [85]. However, ethnic Thais with the *HLA-B48* haplotypes might be susceptible to developing secondary dengue hemorrhagic fever [86], and deletion between the two MIC genes was associated with nasopharyngeal carcinoma in Malaysian Chinese [87] but not Southern Chinese Han [88].

## 3. *HCP5* Single Nucleotide Variant (SNV) Associations with Disease

Many reviews are available that outline the enormous advancements made during the last twenty years to improve our understanding of the MHC locus and genetic susceptibility to autoimmune and infectious diseases because of the availability of dense genotyping platforms and hybridization chips such the custom-made Illumina Infinium SNP chip (Immunochip) [8,89]. The online Genome-Wide Association Studies (GWAS) Catalog [90] revealed that *HCP5* has 131 associations in 86 studies (86 variants) with most severe consequences associated with SNP in the regulatory and transcript region from chromosomal position 31,462,917 to 31,466,072. Some of the different diseases and phenotypes that were associated with the internal and 5′ or 3′UTR single nucleotide variants (SNVs) of the *HCP5* gene, using dense genotyping platforms such as Illumina’s HumanHap550 BeadChips or candidate genotyping by PCR, are listed in Table 3. However, many of these associations need to be considered with caution because they are likely to be part of extended haplotypes or in LD with other identified or unidentified genes in the MHC. In most of these association studies, the *HCP5* SNV was in strong LD with SNV from other MHC gene loci. This is not surprising given the relatively large genetic diversity within the HLA class I genomic blocks observed between different HLA haplotypes [5], which has resulted in the formation of ancestral haplotypes with relatively frozen polymorphic blocks at the HLA class I and class II loci [36].

### 3.1. HIV and AIDS

Fellay et al. [33,34] were the first to find a significant association of the *HCP5* gene variant rs2395029 (T>G) with disease nonprogression in a cohort of HIV-infected individuals. The association was particularly strong with 10% of the variation among individuals during the viral asymptomatic set point (stabilized viral load) period of infection with the minor C/G-allele being associated with a lower viral load. Their hypothesis was that the association observed for the variation in HIV-1 set point was due mainly to *HLA-B*57:01*, although they acknowledged the possible effect of the *HCP5* variation because it was an endogenous retroviral element (ERV) with sequence homology to retroviral *pol* genes, and that its transcripts were known to be expressed in lymphocytes with the possible production of predicted short protein antigens with an amino acid substitution at the SNP rs2395029 [28,30]. A model in which a combined haplotypic effect of *HCP5*, *HLA-B*57:01*, and the SNV rs9264942 located in the 5′ region of the *HLA-C* gene, 35 kb away from transcription initiation and 156 kb telomeric of the *HCP5* gene on the HIV-1 set point, was consistent with the observation that suppression of viremia can be maintained in *B*57:01* patients with undetectable viral load, even if HIV-1 undergoes mutations that allow escape from cytotoxic T lymphocyte (CTL)–mediated restriction. Therefore, *HCP5* seemed to be a good candidate to interact with HIV. Furthermore, Fellay et al. [33] found that the strongest association with progression to AIDS included a set of seven polymorphisms located in and near to the ring finger protein 39 (*RNF39*) and zinc ribbon domain-containing 1 (*ZNRD1*) genes in the MHC regions.

The association of the *HCP5* polymorphism at rs2395029-G with *HLA-B*57:01* and HIV nonprogression to AIDS was confirmed in follow-up studies [93,95]. Limou et al. [95] used their own Genomics of Resistance to Immunodeficiency Virus (GRIV) cohort and the AIDS GWAS of the Euro-CHAVI (Center for HIV/AIDS Vaccine Immunology) cohort and replicated the results of Fellay et al. [33]. As expected, they obtained the strongest association with the *HCP5* rs2395029 minor SNP and the second strongest association with the *C6orf48* rs9368699 SNP, which was in LD with *HCP5* and with several SNPs located in the MHC class I and class III region including *HLA-B*, *MICB, PSORS1C1* (class I region), *TNXB, TNF, LTB, BAT1, BAT2, BAT3*, and *RDBP* (class III region). Their study suggested an independent role for the *ZNRD1* gene in disease progression. Of the 50 best signals found in their meta-analysis, 46 originated from the HLA locus, emphasizing the critical role of the MHC in the control of HIV-1 replication and delayed disease progression.

A GWAS of a multiethnic cohort of HIV-1 controllers and progressors [93] obtained similar results as in the previous two studies [33,95] but with the added novelty that (1) the nature of the HLA-viral peptide interaction was the major factor modulating durable control of HIV infection, (2) *HCP5* rs2395029-G was a proxy not only for *HLA-B***57:01* but also for many protective and risk HLA alleles (predominantly at *HLA-B*), and (3) with an independent effect on *HLA-C* gene expression that together differentially affected the response to HIV and delayed progression to AIDS [106,107]. However, *HCP5* rs2395029 was not associated with viral load at set point in African populations [108], instead, the viral load was associated with the *HLA-B*57:03* allele [92] suggesting a difference in individuals of African and European ancestry. The rs2395029-G polymorphism is missing from the Yoruban population of the Niger-Congo, and they have the *HLA-B*57:03* allele instead of the *HLA-B*57:01* of Europeans [108,109].

The Catano et al. [110] study of mainly European Americans was noteworthy because they showed in their univariate analysis that the *HCP5* minor allele was associated with a slow disease course and lower viral loads, whereas in the multivariate models, after partitioning out the protective effects of *HLA-B*57*, the *HCP5* minor allele was associated with disease acceleration and enhanced viral replication. These contradictory associations for *HCP5* are generally obscured, possibly because of the very strong LD between this allele and a subset of protective *HLA-B*57* alleles. Furthermore, they found that *HCP5* and *HLA-C* alleles stratified the *HLA-B*57*-containing genotypes into those that associated with either disease retardation or progression, “providing one explanation for the long-standing conundrum of why some *HLA-B*57*-carrying individuals are long-term non-progressors, whereas others exhibit progressive disease.” Their study highlighted the strong dependence of genotype–phenotype relationships upon cohort design, phenotype selection, LD patterns, and populations studied. However, in a more recent study [109], the minor alleles of *HCP5* rs2395029, *HLA-C* rs9264942, and *ZNRD1* rs3689068 were associated strongly with lower viral load among antiretroviral-naïve individuals who had a shorter time to first viral load of less than 51 copies/mL during combination antiretroviral therapy, even after adjustment for viral load before combination antiretroviral therapy. The authors of a 2016 study [109] concluded that more studies are needed to elucidate the function and mechanisms of these SNVs in relation to HIV disease progression and disease course as well as to clarify whether these are functional SNV or whether they simply reflect a strong LD with other SNVs that are actually functional but, as yet, unidentified.

In 2010, Yoon et al. [35] infected TZM-bl cells in vitro with HIV-1 and plasmids expressing high levels of *HCP5* transcripts; they showed that the *HCP5* gene had no direct antiviral effect. This implied that the association of an *HCP5* variant with viral control more likely was due to various other complex interactions and epistatic factors. Thus, *HCP5* rs2395029 was deemed more as a useful genetic marker or proxy for *HLA-B*57:01* in Europeans; and it was used as such by other researchers to show that hypersensitivity in HIV patients to abacavir treatment was associated with *HLA-B*57:01* and HIV [97,110,111]. Abacavir is a potent nucleoside reverse transcriptase inhibitor, and it is an integral part of antiretroviral therapy in combination with other antivirals with good efficacy and a favorable long-term toxicity profile in the treatment of HIV. However, serious hypersensitivity to abcavir is recognized as a prohibitive and life-threatening treatment in approximately 8% of Caucasians and 3% of African Americans. Rodriguez-Novoa et al. [97] concluded from their analysis of 245 HIV patients that “the use of HCP5 rs2395029 testing could be as useful as HLA-B*57:01 typing to prevent the abacavir hypersensitivity reaction. Given that HCP5 testing is cheaper, less time-consuming and easier to perform than HLA typing, it may confidently replace the latter in clinical settings.” However, in 2012, when Melis et al. [78] genotyped both the *HCP5* SNP and *HLA-B*57:01* in a set of 1888 samples, they found a good correlation, but they also found that one *HLA-B*57:01*-positive sample tested negative for the *HCP5* SNP, and that *HCP5* could not be amplified in two samples as a consequence of a homozygous deletion of the *HCP5* gene. They concluded that copy number variation and incomplete LD interfered with the *HCP5* genotyping assay for abacavir hypersensitivity, and that ethnicities should be considered when using the *HCP5* SNP as a surrogate marker for *HLA-B*57:01*. Discovering the absence of *HCP5* SNPs in two individuals was not a surprising result because the *HCP5* gene was known to be deleted along with the *MICA* gene in a number of Asian HLA haplotypes including the *HLA-B*48* haplotype in Japanese [42,73]. Despite the occasional deletion or structural mutation, genotyping for the *HCP5* rs2395029 minor allele is still a quick and practical method for assessing the possibility of abacavir hypersensitivity associated with *HLA-B*57:01* in order to avoid potentially fatal consequences [112]. Also, the structurally polymorphic *AluMICB*, which is a lineage indel within intron 1 of the *MICB* gene, possibly could be used as an alternative and simple proxy assay for *HLA-B*57* and *HLA-B*48:01* [79]. Ultimately, genotyping for *HLA-B*57* is essential because the most widely accepted hypothesis proposes that abacavir alters the repertoire of peptides that are able to bind to MHC, allowing for presentation of novel self-peptides, which in the absence of abacavir would not bind to *HLA-B*57:01* [113]. These altered peptides are perceived as foreign by T cells, and a cytotoxic response is triggered that can result in the abacavir hypersensitivity reaction [114].

### 3.2. Herpes Zoster

Crosslin et al. [99] reported that MHC genomic regions of *HCP5*, especially at SNPs rs116062713 (now rs75640364) and rs114864815 (now rs77349273), were strongly associated with susceptibility to herpes zoster (shingles) caused by the varicella zoster virus. The *HCP5* SNP marker rs75640364 is in the 3′UTR of *HCP5* at chr6:31,465,789 and SNP rs77349273 is a 2 kb upstream variant of *HCP5* located between *MICA* and *HLA-X*.

### 3.3. Autoimmune Disease and Drug Hypersensitivity

Apart from HIV infection, AIDS, and abacavir hypersensitivity, the *HCP5* [rs2395029] SNVs also were associated with other adverse drug reactions, autoimmune diseases, and abnormal phenotypes. The same *HCP5* and *HLA-B* genotypes were associated with psoriasis (PS) and psoriatic arthritis (PSA) [96,115] and found to be a major determinant of flucloxacillin-induced liver injury [94]. Liu et al. [96] found that the *HCP5* G2V polymorphism at rs2395029 had the highest odds ratio with both psoriasis (PS) and psoriatic arthritis (PSA) for 223 Caucasian individuals, and its effect was independent (not in significant LD) of the most highly associated SNP rs10484554 that was 34.7 kb upstream of *HLA-C*. The MHC, in particular the HLA class I region, is the only genomic region that has been shown to be consistently associated with PS. The nine top-ranking SNPs in the Liu et al. [96] study were from the MHC, and seven were significant, even when statistically adjusted for multiple testing. The authors also noted that the same significant *HCP5* SNV in psoriasis and HIV infections is not surprising since psoriasis can be triggered by infection with HIV and other viruses. Hence, it is possible that *HCP5*-*C* carriers mount a strong immune reaction to viral infection, and when psoriasis is associated with other genes such as *corneodesmosin, POU5F1, MICA*, and *HLA-C* in the MHC and genes outside the MHC in genetically susceptible individuals [116], then this reaction might lead to excessive inflammation in skin and joints.

The antimicrobial agent flucloxacillin is a common cause of drug-induced liver injury (DILI), but the genetic basis for susceptibility remains unclear. A study of 51 cases of flucloxacillin DILI, along with a replication study of another 23 cases, found that rs2395029(G) carriers were at highly increased risk (odds ratio 80, p = 8.7 × 10^−33^) [94]. The rare homozygote rs2395029(G;G) increases the odds of flucloxacillin -induced liver injury by 45×.

Many associations have been attributed to *HCP5* SNVs outside the gene locus, either upstream in the *HLA-X* pseudogene or beyond the *HCP5* 3′UTR, positioned almost up to the locus for the *HCG* lncRNA. For example, SNP rs3099844 was associated with nevirapine-induced Steven–Johnson Syndrome (SJS), toxic epidermal necrolysis (TEN) drug reactions [117], systemic lupus erythematosus (SLE), anti-Ro/SSA [118], Sjogren Syndrome, leukopenia, and lymphoma [119]. Thus, SNV was referred to incorrectly as being in the *HCP5* gene when it was actually positioned in *LOC102725068* (chr6:31,479,918–31,494,794 of GRCh38/hg38), which was the *MICB-DT* gene that coded for the 14,877 bp *MICB* divergent transcript. The *MICB-DT* lncRNA is interesting in its own right because it immediately neighbors the *MICB* gene, has many transcription factor binding sites, and is associated with various diseases or phenotypes such as myositis, asthma, mumps, plantar warts, blood protein measures, MICB protein levels, and lymphocyte and monocyte counts [90].

In a recent genome-wide association analysis of a total of 915 children with Kawasaki disease and 4553 controls in the Korean population, the susceptibility locus for the disease was identified by Kim et al. [39] to be *NMNAT2* on chromosome 1q25.3 (rs2078087) and the human leukocyte antigen (HLA) region on chromosome 6p21.3 (*HLA-C, HLA-B, MICA*, and *HCP5* with rs9380242, rs9378199, rs9266669, and rs6938467, respectively). The *HCP5* rs6938467 (T/G) is positioned at chr6:31,440,139 (GRCh38.p12) well outside the *HCP5* RNA reference sequence (NR_040662.1) and upstream of the *LINC01149* locus. However, this is within the alternative genomic location of the old Hugo Gene Nomenclature Committee (HGNC definition) of *HCP5* (HGNC ID:21659) using Gencode Gene ENSG00000206337.10 and the Gencode Transcript ENST00000414046.2 starting within the *MICA* gene in the genomic location of chr6:31,400,702–31,465,809, which provided a transcript, including UTRs, of 65,108 bp in sequence length.

### 3.4. Transplantation

*HCP5* upstream and downstream SNVs were associated with disease relapse in 53 patients with unrelated cord blood transplantation (UCBT) when compared to HLA-matched unrelated donors [105]. The diagnosed diseases before transplantation were transfusion-dependent thalassemia (19), genetic diseases (10), anemias (8), leukemias (11), and other neoplastic diseases (5). Of the 58 SNVs that were analyzed by genotyping, seven SNVs were associated with the risk of relapse, and two of these SNVs, rs2523675 and rs2518028, were located ~2.5 kb downstream of *HCP5* but still within the tailing *ERV16* (*ERV3-16A3* and *5′LTR16B2*) sequence of ~4.6 kb. The other SNVs were in the *MICD* gene and the *HLA-DOA* gene. Hematological disease relapses after UCBT were defined as recurrence of malignancy and/or relapse of nonmalignant hematological disorders defined by conversion to partial or complete nonresponse. Although the two *HCP5* SNVs resulted in 2.75 and 4.52 times greater risk of relapse for the recipients than the donors, a possible molecular mechanism for the relapse was not provided.

### 3.5. Cancer

A number of different SNVs located in the *MICA* and *MICB* genes and in the genomic region between them have been associated with cancer. In a recent analysis of eight GWAS datasets with 17,153 cases and 239,337 controls by Yuan et al. [92], at least six *HCP5* SNVs (including rs3130907 within the HCP5 sequence) were associated significantly with lung cancer susceptibility along with the novel risk SNV rs114020893 in the *lncRNA NEXN-AS1* region at 1p31.1. The authors noted that lung cancer risk-related loci (6p21 and 15q25) were enriched in lncRNAs, such as *HCP5*, *RP11-650L12.2*, *XXbac-BPG27H4.8*, and *HCG17*, and that using noncoding regions in GWAS and gene-based and pathway-based analyses should be complementary to protein coding-related approaches [92].

## 4. *HCP5* Methylomics

The study of differentially methylated sites, differential gene expression, and epigenetic mechanisms represent a complementary method to genetic association studies for the identification of molecular and biological pathways that contribute to good health during a normal life cycle and to clinical heterogeneity of autoimmune and chronic diseases and cancer [120]. Recent studies have identified differentially methylated sites within, or neighboring, the *HCP5* gene sequence associated with epigenetic regulation of various disease phenotypes (obesity, SLE) and in response to fetal development, aging, HIV infection, and vaccination (Table 4).

Hypomethylation of *HCP5* was associated with autoantibody production against dsDNA, Sjogren’s syndrome-related antigen A (SSA), Smith (Sm) antigen, and ribonucleoprotein (RNP) in SLE [122,126], with gene expression and humoral immune response to influenza [124], with hypomethylated PSORS1C1-associated allopurinol-induced severe cutaneous adverse reactions in Han Chinese [127], accelerated aging in chronic HIV infection [123], endometrial receptivity [125], sexual bias in the human placental sexome [128,129], age-related monocyte and T cell gene expression [121], lung adenocarcinoma [130], and with hypomethylated *POU5F1*-associated ankylosing spondylitis [131]. In contrast to the more common observation of hypomethylation, HCP5 hypermethylation was associated with obesity and BMI in an epigenome-wide association study of adiposity in Ghanaian African migrants using whole blood to measure DNA methylation [64].

Age-related HCP5 DNA methylation was associated with gene expression in human monocytes and T cells [121], and the expressed genes that linked to potentially functional age-related methylation sites were enriched with antigen processing and presentation MHC class I and class II genes that were implicated in ‘parainflammation’ and the development of age-related chronic inflammatory diseases and autoimmune diseases. The total effects of age on gene expression (which increased with age) were significant (FDR < 0.05) for seven MHC genes—*HLA-B, -E, -DPA1, -DPB1, TAP2, TAPBP*, and *HCP5*—with hypomethylation within and/or near to all of those genes. On the other hand, Gross et al. [123] found an HCP5 CpG DNA methylation signature in blood cells of patients with chronic, well-controlled HIV infection that correlated with accelerated aging, and that it also was independently associated with HLA expression and corresponding HIV control. The level of methylation at HCP5 was correlated with a patient’s CD4^+^/CD8^+^ T cell ratio to provide further evidence that the observed changes were functional [123]. Chronic HIV infection, even when viral loads were kept below the level of detection, is associated with early onset of diseases linked to aging, including cardiovascular disease, kidney disease, cancer, and premature death. Highly active antiretroviral therapy (HAART) controls the burden of HIV, without curing the infection, enabling HIV-infected patients to live for many decades, provided they continue their medications. The increased methylation changes in HIV-infected patients found beyond their chronological age suggested about a five-year increase in aging compared to healthy controls [123].

Systemic lupus erythematosus (SLE) is a chronic inflammatory autoimmune disease of unknown etiology that can affect most organs and is characterized by the development of autoantibodies associated with specific clinical manifestations implicated in the pathogenesis of lupus nephritis and decreased survival [122,126]. The genetic risk factors suggested for SLE include alleles in *IRF5, STAT4, BLK, TNFAIP3, TNIP1*, *FCGR2B*, and other genes [132]. Genome-wide DNA methylation analysis of SLE revealed persistent hypomethylation of interferon genes and compositional changes to CD4^+^ T cell populations. For example, Chung et al. [122] characterized the methylation status of 467,314 CpG sites in 326 women with SLE DNA methylation profiling, performed using the Infinium HumanMethylation450 BeadChip (Illumina), and they identified and replicated significant associations between anti-dsDNA autoantibody production and the methylation status of 16 CpG sites in 11 genes. Differential methylation for these CpG sites was also associated with anti-SSA, anti-Sm, and anti-RNP autoantibody production. Overall, associated CpG sites were hypomethylated in autoantibody-positive samples compared to autoantibody-negative cases. In the discovery/replication analysis, associations with hypomethylated CpG sites were within genes (*IFIT1, IFI44L, MX1, RSAD2, OAS1*, and *EIF2AK2*) that were either induced by type 1 interferon or that regulated type 1 interferon signaling (NLRC5). Except for hypomethylation at *HCP5* and the *PSMB8* gene in the class III region, differential methylation of CpG sites within the MHC was not strongly associated with autoantibody production. Thus, hypomethylation of CpG sites within *HCP5* and other genes from different pathways that could not be explained by DNA sequence variation were associated strongly with anti-dsDNA, anti-SSA, anti-Sm, and anti-RNP production in SLE.

In a system-wide association study between DNA methylation, gene expression, and humoral immune response to influenza [124], a cohort of 158 individuals who were 50 to 70 years old showed that *HCP5* along with *HLA-B* and *HLA-DQB2* had an important role in methylation expression, particularly when the humoral immune response to influenza was measured by a hemagglutination inhibition assay (HAI). Only two genes showed association in all three independent analyses: *ADARB2*, an inhibitor of adenosine deaminase activity (RNA editing), and *SPEG*, a kinase with a known function in myocyte development. The small number of genes, including *HCP5*, that overlapped across two or more methods and at multiple time points were *HLA-B*, *HLA-DQB2*, the histone deacetylase *HDAC4, RWDD2B, PTPRN2* that (de)phosphorylates phosphoinositols in an insulin regulatory role, *DNAH2, HCP5, FAM24B, LOC399815*, and the transcription factor genes *PAX7* and *PAX9*. Many genes (~640 genes) that were identified in one of these analyses had direct protein–protein interacting genes identified in the other analyses, revealing that the impact of methylation on humoral immunity is complex and highly dependent upon the immune outcome. Zimmermann et al. [124] reported that methylation levels of a CpG within the gene body of *HLA-B* (hypermethylation) were strongly associated with HAI and had an opposite trend to that of *HCP5* (hypomethylation).

Human and animal studies have identified that the placenta expresses select transcripts in a sexually dimorphic manner [129]. A microarray-based study identified sex-dependent differences in the placental transcriptomic profile in males and females (sexome) with isolated cells derived from human placental villi [128]. The four cell types examined included cytotrophoblasts, synctiotrophoblasts, and arterial and venous endothelial cells. For sex-dependent differences, the males demonstrated enrichment of signaling pathways previously reported to mediate graft versus host disease and transcripts involved in immune function and inflammation such as *HLA-DQB1* (syncytiotrophoblast), *HLA-DQA1* (syncytiotrophoblast and cytotrophoblast), *HCP5* (cytotrophoblast), *NOS1* (cytotrophoblast), *FSTL3, PAPPA, SPARCL*, *FCGR2C* (trophoblast epithelium), *CD34* (cytotrophoblast), *HLA-F* (cytotrophoblast), and *BCL2* (syncytiotrophoblast). Males demonstrated a greater in utero vulnerability, and the findings of Cvitic et al. [128] suggested that these effects may partially be due to reduced maternal–fetal compatibility for males who were then required to up-regulate immune-associated transcripts in an attempt to combat an attack by the maternal immune system.

In a genome-wide methylome analysis of endometrial biopsies collected from 17 healthy fertile-aged women from prereceptive and receptive phases of a menstrual cycle, Kukushkina et al. [125] found that extracellular matrix organization and immune response were the pathways most affected by methylation changes during the transition from prereceptive to receptive phase. The overall methylome remained relatively stable during the two time points of the menstrual cycle with small-scale changes affecting 5% of the studied CpG sites (22,272 out of 437,022 CpGs, FDR < 0.05). The study confirmed that the differential methylation of *KRTAP17-1, CASP8, RANBP3L, WT1, MPP7, PTPRN2*, and *HCP5* between the early and midsecretory phases were similar to those observed in the previous studies [133,134]. The differential methylation of *PTPRN2* and *HCP5* in the endometrium is an interesting connection, given that they both were differentially methylated in a system-wide association study between DNA methylation, gene expression, and humoral immune response to influenza [134]. The *PTPRN2* (NCBI gene: 5799) gene product (de)phosphorylates phosphoinositols, has an insulin regulatory role, and it may be an autoantigen in insulin-dependent diabetes mellitus, but its actual function as a methylation site in the endometrium or human monocytes and T cells is not known.

In an epigenome-wide association study using whole blood measures of adiposity in 547 Ghanaian African migrants, Meeks et al. [64] found that obesity and body mass index (BMI) were related to *HCP5* hypermethylation and 18 differentially methylated positions (DMPs) for BMI, 23 for waist circumference, and three for obesity. Fourteen DMP overlapped between BMI and waist circumference. The two epigenome-wide loci that were significantly hypomethylated for both general adiposity and abdominal adiposity were *CPT1A* (carnitine palmitoyltransferase 1A) and *BCAT1* (branched chain amino acid transaminase 1), whereas *NLRC5* ((NLR family CARD domain containing 5) and most other DMPs including those for six HLA genes—*HCP5, HLA-B, TAP1, TAP2, PSMB8*, and *HLA-E*—were hypermethylated. The hypermethylation of *NLRC5* was highly significant, and this gene is known to regulate the expression of MHC class I genes and to limit the activation of inflammatory pathways [54,55,56]. Thus, the results of Meeks et al. [64] suggested that obesity might suppress the adaptive immune response and induce inflammation that could also result in insulin resistance.

Coit et al. [131] identified a total of 68 differentially methylated sites between ankylosing spondylitis (AS) patients and osteoarthritis controls. *HCP5* and *POU5F1* were both hypomethylated in *HLA-B*27*-positive compared to *HLA-B*27*-negative AS patients. They suggested that *HLA-B*27* might play a role in AS in part through epigenetic linkage disequilibrium-inducing epigenetic dysregulation. The *POU5F1* gene (alias *OCT4*) is located at the telomeric end of the MHC beta-block, ~98 kb upstream of the *HLA-C* gene, and its role in methylation is well described [120].

*HCP5* is known to be involved in lung cancer [130], and Yuan et al. [92] described at least six *HCP5* SNVs, including rs3130907, that were associated significantly with lung cancer susceptibility (Table 3). Previously, Orvis et al. [135] showed that inactivation of the *BRG1* gene, also known as *SMARCA4*, which encodes the ATPase subunits of the SW1/SNF chromatin remodeling complex, contributed to non-small cell lung cancer aggressiveness by altering nucleosome positioning in a wide range of genes as well as by downregulating the expression of *HCP5* and all of the classical and nonclassical HLA class I genes.

Presumably, hypomethylation of *HCP5* leads to added interactions and connectivity with proteins and other RNA sequences, especially with miRNA regulators. Studies on hypermethylation of *HCP5* are still lacking, and such studies might provide a more contrasting view of the action of methylation on the function of *HCP5* in health and disease. However, the overall absence of hypermethylation data for *HCP5* might be related to the DNA methylation paradox, whereby methylation of the transcribed region and the region of transcription initiation have opposite effects on gene expression [120]. Although methylation can affect gene expression in both directions depending on the genomic region, there are more negative correlations in the 5′ UTR, while positive correlations are more common in the gene body region. While this was the case for *HLA-B*, the reverse was observed for *HCP5* in humoral immune response to influenza [124].

## 5. *HCP5*, Gene Targets, and Transcription Factors in Interaction Networks

Many hundreds of different transcription factors (TFs) are believed to target the *HCP5* sequence and regulate its expression. Although only a few experiments have examined the relationship between particular transcription factors and the expression of *HCP5* and its neighboring genes in the region between *MICA* and *MICB*, a variety of datasets have predicted connections between *HCP5* and many known TFs. Appendix A shows five Internet databases sourced from Harmonizome and GeneCards (Appendix A) that associated TFs with *HCP5*. For example, the dataset of JASPAR Predicted Transcription Factor Targets predicted that 57 transcription factors were associated with regulating the expression of *HCP5*, whereas the MotifMap Predicted Transcription Factor Targets dataset predicted only seven associations: alpha-CP1, E2A, ETS2, MAFA, NF-kB, NF-Y, and TEF-1. On the other hand, the TRANSFAC curated dataset only listed PTF1A as a TF, interacting with the *HCP5* gene in low- or high-throughput transcription factor functional studies. PTF1A is the pancreas-specific transcription factor 1a, with a role in mammalian pancreatic development and in determining whether cells allocated to the pancreatic buds continue towards pancreatic organogenesis or revert back to duodenal fates. Also, *HCP5* is only one of 233 target genes for the PTF1A transcription factor. In contrast, the TRANSFAC dataset predicted that 13 TFs regulated the expression of the *HCP5* gene: ATF2, ELF3, ETS1, HINFP, JDP2, LEF1, LTF, MYC, NFE2L2, RUNX1, SMAD4, SMARCA2, and SPI1. Most of these predicted transcription factors also targeted many other genes as part of interaction networks, cascades, or divergent pathways.

In a study of interactions between *HCP5* and transcription factors, Warner et al. [136] showed that *HCP5*, together with *NOD2* and *IL-8*, was associated strongly with decreased viability of cells in a study of the inactivation of the *NF-kB1* gene by knockout. In their datafile, the other MHC genes that were strongly associated with decreased viability were *HLA-A, -DQA1, -DRB1, -DRB4, -DOA*, and -*DOB* but not *HLA-B, -E, -G, -F, -DRB5, -DMA, -DRA –DQA2, -DQB1, -DRB3, -DMB*, -*DPB1, -DPA1, MICA, MICB, TNF, LTA, LTB, C4A*, and *C4B* in the HEK293 cell line. Thus, NF-kB1 (located on chr 4) regulates *HCP5* and the gene expression of some other MHC genes as well as a wide variety of biological functions, including inappropriate activation associated with inflammatory diseases, inappropriate immune cell development, and delayed cell growth. This study [136] also demonstrated that although *HCP5* was often in LD with many genes in the MHC, it could be activated or suppressed independently from most of them.

Coit et al. [131] identified a total of 68 differentially methylated sites in a study of ankylosing spondylitis (AS) patients and osteoarthritis controls; *HCP5* and *POU5F1* were both hypomethylated in *HLA-B*27*-positive compared to *HLA-B*27*-negative AS patients. This predicted cis relationship between the transcription factor *POU5F1* and *HCP5* is interesting, given that both genes are located in the beta block of the HLA class I region [1,2]. Also, Meeks et al. [64] in an epigenome-wide association study of measures of adiposity among Ghanaians showed that *NLRC5* and *HCP5, HLA-B, TAP1, TAP2*, *PSMB8*, and *HLA-E* were all significantly hypermethylated for both general adiposity and abdominal adiposity.

The interactive website Pathwaynet [137] predicts both the presence of a functional association and the most likely interaction type among human genes or their protein products on a whole-genome scale. It is based on a large compendium of refined regulatory interactions within 77 tissues, with their curated pathways taken from primary experimental datasets such as 690 ChIP-Seq datasets, numerous mass spectrometry of metabolites, protein–protein interactions, disease samples, etc., in order to capture the interaction networks. Figure 5 shows the top 15 genes that interacted with *HCP5*, as predicted by Pathwaynet [137], with a high relationship confidence of between 0.9247 (*BTN3A3* and *IRF9*) and 0.9523 (*HLA-F*). All of the genes were within the MHC region except for *UBE2L6* (chr11q12.1), *TRIM22* (chr11p15.4), *IRF1* (chr5q31.1), *CASP1* (chr11q22.3), and *IRF9* (chr14q12). Pathwaynet did not specify the type of functional relationships between *HCP5* and these 15 genes or gene products; therefore, the predicted gene interactions should be considered with considerable caution. However, it is evident that the non-HCP5 genes have roles in antigen processing and presentation, the proteasome, graft versus host disease, allograft rejection, autoimmune disease, response to type 1 interferon or interferon gamma, regulation of viral reproduction, IL-6- and IL-12-mediated and NOD-like signaling pathways, and signal transduction by the p53 class I mediator. This connection is supported to a large degree by the top 52 genes that were positively associated with *HCP5* gene expression in the Comparative Toxicogenomics Database (CTD) datasets (Table 5). The *HCP5* interaction and regulation is probably by way of the methylome and the competitive endogenous RNA regulatory networks, although this premise needs to be investigated further in both in vivo and in vitro experiments and association studies.

## 6. *HCP5* Gene Expression and Gene Interactions

In 1993, Vernet et al. [27] originally reported that *HCP5* expressed a 2.5 kb transcript in human B cells, phytohemagglutinin-activated lymphocytes, a natural killer-like cell line, normal spleen, hepatocellular carcinoma, neuroblastoma, and other nonlymphoid tissue but not in T cells. Since then, numerous studies of genome-wide gene expression using dense Affymetrix expression arrays were published, but the findings rarely reported directly on the expression of *HCP5*. To identify the expression profile of *HCP5* in various scenarios, the databanks needed to be investigated and interrogated separately. In this way, it was possible to find the particular pattern of *HCP5* expression especially in comparison to the other class I and class II genes. *HCP5* is widely expressed at low levels, but it is primarily expressed at higher levels in cells of the immune system such as spleen, blood, and thymus (http://smd-www.stanford.edu/), consistent with potential roles in autoimmunity and cancer.

Harmonizome (Appendix A) was a good starting point to find data about *HCP5* expression under various experimental conditions [138]. The gene expression results of *HCP5* in 53 tissues from 8555 samples (570 donors) were sourced from GTEx RNA-seq using the University of California, Santa Cruz (UCSC) online browser (Appendix A). Also, interrogation of the online NCBI Gene Expression Omnibus (GEO) with the keyword “HCP5” produced 1771 results to review. Eighty-nine results were related to up and down differential expression, 6 results to the keyword ‘immunity’, 375 results to ‘cancer’, 28 to ‘HIV’, 50 to ‘virus’, 36 to ‘interferon’, 42 to ‘host defense’, and 4 to ‘MHC’. These results were browsed with a visual profile of the effects of treatments and experiments on the gene expression of *HCP5* and/or other genes of investigative choice. Nine studies in GEO confirmed that IFN and IL28B upregulated *HCP5* RNA in some cell types, whereas IL10 downregulated *HCP5* RNA in peripheral blood mononuclear cells (PBMCs) (Table 2). There were little or no significant data for other cytokine-positive or -negative effects on *HCP5*.

In comparison, Appendix A shows the ~37 drugs and chemicals that induce or suppress *HCP5* expression with effects on inferred diseases, and that were identified in the Comparative Toxicogenomics Database (CTD) [139]. Based on the data in Appendix A, *HCP5* gene expression seems to be decreased by various immunosuppressants and neurotoxins. This includes the immunosuppressive aflatoxin B1 that increased methylation of the *HCP5* gene [140]. In addition, the CTD database [139] revealed that *HCP5* reacted with 714 different genes in various gene expression studies. The top 52 genes that *HCP5* interacted with most often included *PTGS2, TNF, IL1B, PTGS1*, and *CASP3* (Table 5). The summary information provided by NCBI RefSeq for each of these five genes was the following: PTGS1 and PTGS2 are prostaglandin-endoperoxide synthases or cyclooxygenases, key enzymes in the biosynthesis of prostaglandin that are regulated by specific stimulatory events involved in inflammation and mitogenesis. The tumor necrosis factor (*TNF*) gene that is located downstream of *HCP5* in the MHC class III region encodes a multifunctional, proinflammatory cytokine that is a member of the tumor necrosis factor (TNF) superfamily. This cytokine is mainly secreted by macrophages. It is involved in the regulation of a wide spectrum of biological processes including cell proliferation, differentiation, apoptosis, lipid metabolism, and coagulation. It has been implicated in a variety of diseases including autoimmune diseases, insulin resistance, and cancer. Knockout studies in mice suggested that TNF also has a neuroprotective function. IL1B is a cytokine expressed on chromosome two and produced by activated macrophages as a proprotein, which is proteolytically processed to an active form by caspase 1 (CASP1/ICE). This cytokine is an important mediator of the inflammatory response and is involved in a variety of cellular activities including cell proliferation, differentiation, and apoptosis. IL1B induces PTGS2/COX2 in the central nervous system and contributes to inflammatory pain hypersensitivity. CASP3 or caspase 3 is a protease with a central role in the execution phase of cell apoptosis. It inactivates poly(ADP-ribose) polymerase, while it cleaves and activates sterol regulatory element binding proteins as well as caspases 6, 7, and 9. Also, it is the predominant caspase involved in the cleavage of amyloid-beta 4A precursor protein, which is associated with neuronal death in Alzheimer’s disease. Therefore, it is evident that *HCP5* RNA is strongly associated with the inflammatory innate immune response as well as adaptive immune responses as indicated by its coexpression with various class I genes in expression studies in the databases (e.g., GEO) and the published literature.

### 6.1. HCP5 Expression in HIV-Infected Cells

Given that *HCP5* has been associated strongly with viral suppression in HIV-infected cells in GWAS (Table 3), it is surprising that so few papers have specifically addressed the correlation between *HCP5* gene expression levels and HIV levels or response to HIV infection [35]. However, there are a few studies in the GEO database to suggest that *HCP5* expression in response to HIV is induced, suppressed, or unaffected in some cell types. For example, *HCP5* RNA was significantly lower in the three HIV-negative controls than in three samples of jejunal mucosal cells from HIV patients on highly active antiviral therapy [141]. Also, *HCP5* transcription activity was high in three mononuclear cell samples, but it was low or absent in three T cell samples and three fibroblast samples with HIV DNA integration sites [142]. Similarly, *HCP5* RNA was higher in the brains of 10 of 26 patients receiving antiretroviral therapy for HIV-associated neurocognitive disorder than in nine uninfected controls [143]. However, in some studies, there was no significant effect of HIV on *HCP5* RNA levels. For example, there was no difference in the *HCP5* RNA levels of 23 infected and 12 noninfected peripheral blood mononuclear cell samples [144], little difference between eight infected and eight uninfected macrophage samples [145], and little or no difference between five uninfected and 15 HIV-infected CD4^+^ samples and 15 CD8+ T cell samples [146]. Alternatively, *HCP5* RNA was low or absent in three T cell samples infected with HIV-based vector or three samples treated with TNF-alpha, but it was relatively higher in the three untreated T cell samples [147]. Unfortunately, in the one study on RUNX1 in the regulation of HIV, no data were provided about *HCP5* RNA levels. Thus, based on these limited analyses, the role of *HCP5* in HIV infection and AIDS remains unclear.

### 6.2. HCP5 Expression in Cancer

*HCP5* has been found upregulated or downregulated in a number of different cancers. The interactions between *HCP5* and three transcription factors with potential antioncogenic functions are noteworthy. *HCP5* was confirmed as one of the *KAT8* (alias hMOF) downregulated genes by qPCR and ChIP in the hMOF siRNA knockdown HeLa cells and 20 of 28 clinically diagnosed ovarian cancer tissues [148]. KAT8 (lysine acetyltransferase 8) encodes a member of the MYST histone acetylase protein family involved with the p53 pathway and chromatin organization as well as with the suppression of epithelial to mesenchymal transition and tumor progression [149]. In contrast, Teng et al. [150] demonstrated that the *HCP5* transcribed sequence interacted with an miRNA sequence and the runt-related transcriptional regulator RUNX1 in a feedback loop to regulate the malignant behavior of glioma cells of the brain. Another noteworthy interaction was between *HCP5* and SATB1 (special AT-rich sequence binding protein 1), which is a nuclear matrix-associated DNA binding protein that functions as a chromatin organizer. SATB1 is highly expressed in aggressive breast cancer cells and promotes growth and metastasis by reprogramming gene expression [151]. It also enhanced *HCP5* epigenetically and suppressed the oncogenic long noncoding RNA urothelial carcinoma-associated 1 (*UCA1*) in breast cancer cells. Recently, Zhao and Li [57] showed that transcription factor SP1 induced upregulation of *HCP5*, which in turn promoted the development of osteosarcoma, whereas inhibition of *HCP5* expression reversed cell invasion and epithelial–mesenchymal transition. In addition, *HCP5* is overexpressed in tumor tissues of patients with lung adenocarcinoma, and it is positively correlated with poor prognosis specifically in patients who are smokers with *EGFR* and *KRAS* mutations [56]. *HCP5* also was overexpressed in lymph node metastasis of small cell lung cancer [130,152], glioma tissue [150], colorectal cancerous tissue [153], and cancers of the colon [154], thyroid [155], cervix [156], and breast [151,157].

Interrogation of the TCNG Cancer Network Galaxy Database (Appendix A) produced 206 networks for genes regulating or regulated by the *HCP5* gene as estimated from publicly available cancer gene expression data. There are ~1010 genes that were predicted to interact with *HCP5* either as a child node (regulated gene) or parent node (regulating gene). In about 590 interactions, *HCP5* was the parent or regulatory node, and in the remainder (420 nodes) *HCP5* was the child or regulated node. For example, *HLA-A, -B, -G, -H, and –J* were ranked as child nodes downstream of the *HCP5* parent node in four experimental arrays on breast cancer and one experiment on colon cancer. That is, *HCP5* was predicted to regulate the HLA-class I genes in those experiments. On the other hand, *HCP5* was predicted to be the regulated or child node with *NLRC5*, a member of the NOD-like receptor family that acts as a transcriptional activator of MHC class I genes [61,62,63], that was the parent or regulatory node in some gene expression experiments such as between adenocarcinoma and squamous cell carcinoma in non-small-cell lung carcinoma, breast cancer cell line profiles, non-Hodgkins lymphoma cell lines, complex genetic sarcomas, meningiomas, prostate cancer, and uveal melanoma primary tumors. Since there were far too many data to review here, further information on the activation or suppression of *HCP5* gene expression in many different cancers can be obtained by interrogating the TCNG Cancer Network Galaxy Database with ‘HCP5’ as the search query and following the links including those to the expression arrays at GEO. A more detailed account of *HCP5* RNA interaction in micro RNA regulatory networks in cancer is provided in the following section.

## 7. *HCP5* lncRNA Interactions with Regulatory miRNA in Cancer

In recent years, an increasing number of lncRNA, including *HCP5*, were found to have potential functions in cancer [158,159,160]. The oncogenic lncRNA appear to regulate the transcription and translation of neighboring and distant genes by cis and trans-regulatory functions in a series of biological steps involving dosage compensation, genomic imprinting, and cell cycle dysregulation leading to cancer and its progression [158,159,160]. A particularly important mechanism to emerge from many of these studies is the role of lncRNAs to bind with regulatory miRNA that control antioncogenic or oncogenic pathways. This three-way binding interaction between lncRNA, miRNA, and regulatory protein coding genes, such as those coding for regulatory transcription factors, has become known as the competing endogenous RNA (ceRNA) mechanism/network [161,162]. In this regard, association studies (Table 3), expression data analysis, and knockdown experiments (Table 2, Table 4, Table 5 and Table 6, Appendix A) have shown that *HCP5* can promote or suppress cancers depending on the *HCP5* allelic form and the cancer type. Since 2016, at least ten different cancer types were found to occur and/or progress by way of the *HCP5*–miRNA–gene regulator interactions or the ceRNA mechanism (Table 6).

In an integrated analysis on the dosage effect of lncRNAs in lung adenocarcinoma, Wei et al. [130] found that the protein coding genes *CTSS, FGL2*, and *PDCD1LG2* (alias *PDL2*) competed with *HCP5* (ENSG00000206337) and formed a single regulatory subnet with the miRNAs miR-106b-5p and miR-17-5b. This in part confirmed a previous study that found that *HCP5* was involved in the process of lung cancer by competing with *PDL2*, an immune checkpoint gene, and *FGL2*, a therapeutic target to suppress carcinogenesis [92]. This also was consistent with their finding that at least six *HCP5* SNPs, including rs3130907, were associated significantly with lung cancer susceptibility along with the novel risk SNP rs114020893 in the *lncRNA NEXN-AS1* region at 1p31.1.

*HCP5* expression was positively correlated with the oncogenesis of a pathological grade of glioma tissues, and knockdown of *HCP5* exerted tumor-suppressive effects in human glioma cells by allowing an increase in expression of the miRNA tumor suppressor miR-139 [150]. The malignant behavior of glioma cells in the brain appears to be regulated by an *HCP5*–miR-139–*RUNX1* feedback loop, whereby *RUNX1* increased the promoter activities and expression of *HCP5* that binds to the tumor suppressor miRNA-139 (miR-139) and, therefore, acts as an oncogene. *HCP5* absorbed the tumor suppressor miR-139 to downregulate its expression. Upregulated *RUNX1* also inhibited apoptosis in glioma cell-lines U87 and U251. *RUNX1* down-regulation by knockdown using miR-139 as an inhibitor had the opposite effect on apoptosis and exerted tumor-suppressive effects in human glioma cells. MiR-139 inhibited *RUNX1* expression by targeting the 3′-UTR, and *HCP5* knockdown suppressed *RUNX1* expression by allowing miR-139 overexpression [150]. Thus, it was concluded that *HCP5* promotes cell proliferation, cell migration, and invasion (motility) and inhibits apoptosis, as do many other lncRNAs participating in glioma phenotypes [165].

*RUNX1* also is known to suppress HIV reactivation in T cells, resulting in a negative correlation between *RUNX1* expression and viral load. The pharmacologic inhibition of *RUNX1* by a small molecule inhibitor, Ro5-3335, synergized with the histone deacetylase (HDAC) inhibitor SAHA (Vorinostat), enhanced the activation of latent HIV-1 in cell lines and PBMCs from patients [166]. However, the effect of *HCP5* on *RUNX1* expression in HIV-infected cells is not known, although this effect might occur by way of epigenetic or transcriptomic regulation.

Liang et al. [155] found that lncRNA *HCP5* promoted follicular thyroid carcinoma (FTC) progression as a competing endogenous RNA (ceRNA) sponge for miR-22-3p, mi-186-5p, and miR-216a-5p and activated alpha-2,6-sialyltransferase 2 (*ST6GAL2*). Functional experiments showed that *HCP5* promoted *ST6GAL2*, which in turn mediated the proliferation, migration, invasiveness, and angiogenic ability of FTC cells. In comparison, Yu et al. [156] showed that overexpressed *HCP5* promoted the development of cervical cancer by absorbing miRNA-15a to promote expression of the MET transcription regulator MACC1 (*MACC1*). MiRNA-15a overexpression in vitro inhibited *MACC1* expression and suppressed the proliferation of cervical cancer cells. In contrast, miRNA-15a knockdown experiments or absorption by *HCP5* allowed increased *MACC1* expression and the proliferation of cervical cancer cells. Furthermore, *HCP5* and *MACC1* were overexpressed in cervical cancer tissues compared to paracancerous tissue, and the survival rate of patients with cervical cancer was negatively correlated to *HCP5* expression and positively correlated to miRNA-15a. Luciferase reporter gene assay also showed that miRNA-15a bound directly to either *HCP5* or *MACC1*. Because HPV16 and HPV18 are associated strongly with cervical cancer progression, the question arises whether the *PMSP* gene [38] of *HCP5* is also expressed and what role it might have with the increased proliferation of cervical cancer cells.

Both *MACC1* and *HCP5* are expressed in gastric cancer, and their expression may be regulated by miRNAs. In a metabolic network analysis, Mo et al. [163] found that *HCP5* was coexpressed with 34 metabolic-related protein-coding genes and five lncRNAs, and regulated by three miRNAs, miR-128, miR-101, and miR-103a, that were downregulated in gastric cancer. The *TOPORS-AS1* lncRNA and the NADH ubiquinone oxireductase subunit B6 (*NDUFB6*) coding gene that were associated with *HCP5* in the network analysis were downregulated in gastric cancer samples [163]. In some cancers, TGFbeta might induce *HCP5* transcription via the activity of SMAD3 [56], and increased levels of *HCP5* RNA might either directly or indirectly affect *GSR*, *ASCL1*, *MET*, *GRM8*, *DACHI* [152], *ETN3A1*, *ETN3A3, CCDC50, HERC6, TAP1*, and *PSMB9* [164] as well as many other genes (Table 6).

Based on a data expression analysis, Olgun et al. [157] found that *HCP5* was one of seven lncRNA that interacted with miRNA in breast cancer. *HCP5* bound to miR-155 at the hub of the ceRNA network of interactions in the basal subtype of breast cancer. This basal subtype in breast cancer was characterized by a positive correlation between immune cell infiltration and aggressiveness with a key role for interferon signaling and the induction of cell proliferation by the complement cascade. Olgun et al. [157] detected *C2, C3, C3AR1, C4A*, and *C7* complement genes in the basal ceRNA interactions, suggesting that the complement cascade pathway may be significant for progression of the basal subtype.

In an analysis of differentially expressed profiles of lncRNAs and mRNA in ceRNA networks during transformation of diffuse large B cell lymphoma (DLBCL), Tian et al. [164] identified *HCP5* as a key regulator interacting with many miRNA and protein coding genes associated with transcription (*KLF2*), cell adhesion and proliferation (*CD47*), lipid metabolism (*BTN3A1*), and the adaptive immune response (*TAP1, PSMB9*). However, they concluded that the molecular function of *HCP5* remained unknown.

*HCP5* has been associated with various other cancers including cutaneous melanoma [167], HPV-infected head and neck squamous cell carcinoma [168], squamous cell carcinoma cells [169], HCV-induced liver cancer [170], and upper tract urothelial carcinoma [171] in which ceRNA cross-talk has yet to be tested. The ceRNA network and cross-talk mechanism also might have a role in autoimmune diseases such as idiopathic thrombocytopenia [172], viral infections, and *HCP5*-associated phenotypes (Table 3, Table 4 and Table 6) that, as yet, have not been examined for interactions with miRNA and with the other lncRNA regulators and protein coding genes.

## 8. Perspective

Since the discovery of *HCP5* in 1993 [27], a large amount of data has been gathered about its expression, function, and disease associations. Much of this data, however, is buried in large datasets that require considerable effort to locate, analyze, validate, and interpret [89,90,137,138,139] (Appendix A). Nevertheless, sufficient amounts of published and unpublished data that were retrieved from online public databanks for this review revealed that *HCP5* had important roles in health and disease, particularly with respect to its role as a ceRNA regulator and biomarker in autoimmune diseases and cancer. GWAS indicated that the *HCP5* SNV rs2395029 was a potential marker for abacavir-induced hypersensitivity, a marker for *HLA-B*57:01* in populations of mainly Caucasian or Hispanic descent, as well as flucloxacillin drug liver injury, HIV control, and psoriasis and psoriatic arthritis (Table 3). Other SNVs within the *HCP5* gene or within 2.5 kb of the 5′ or 3′ UTR region are useful association markers for various diseases including myositis, herpes infection, cancer, and risk of relapse after transplantation (Table 3)**.** Methylomic studies have associated *HCP5* strongly with HIV progression, SLE, ankylosing spondylitis (AS) and obesity. Various expression studies have shown that *HCP5* is a useful biomarker for interferon and IL28-related inflammatory response, monocyte response to influenza A infection, and various cancers (Table 4, Table 6, and Appendix A). Many of the *HCP5*-associated diseases such as SLE, AS, psoriasis and psoriatic arthritis, myositis, obesity, and cancer are associated also with accelerated aging, morbidity, and mortality.

The *HCP5* gene within the MHC class I genomic region has evolved by exaptation from an ancient endogenous retroviral and gained a new regulatory function by sequestering the MHC promoter and enhancer region from a fragmented/deleted ancient HLA class I gene. It appears to have generated the *PMSP* sequence that codes for a 61 aa peptide that binds to the HPV L2 capsid protein for viral assembly [41]. The function of the PMSP peptide in different cell types other than its interaction with HPV in keratinocytes still needs to be determined and whether the *PMSP* gene is regulated or continuously expressed along with the *HCP5* transcript. The *HCP5* promoter also contains a 22 nt *RUNX1* sequence (Appendix A) that might contribute to the interaction between the *HCP5* transcript and the Runx transcription factor in glioma and in the monocytes of HIV patients. Because the *HCP5* promoter has many of the canonical TF binding sites of the HLA class I promoter (Appendix A), *HCP5* is often expressed concomitantly with classical and nonclassical class I genes, unless they are differentially separated from each other by epigenomic or ceRNA regulatory processes. For example, in some instances, *HCP5* might be upregulated, whereas class I HLA genes are downregulated as exemplified in the study of humoral immune response to influenza [124]. While it is notable that *HCP5* is often in LD with the HLA class I genes and that their expression is often coordinated in response to various stimulators or suppressors, functional interactions between the products of these genes suggest that *HCP5* has an associated role in antigen processing and presentation, the proteasome, graft versus host disease, allograft rejection, autoimmune disease, response to type 1 interferon or interferon gamma, and in the regulation of viral reproduction, especially in HIV restriction [6,14,24,50,52,107]. The mechanisms by which HLA class I genes and the lncRNA genes *HCP5* and *ZNRD1* might interact with HIV also are worth investigating further because these lncRNAs could be exploited therapeutically using small RNA inhibitors [109,110]. Similarly, the interaction of *HCP5*, microRNA, and protein coding genes in cancer (Table 6) suggests that *HCP5* could be targeted for knockdown or knockout in antitumor therapeutics.

It is evident from the results and reports presented in this review that *HCP5* contributes to regulating viral and autoimmune diseases and cancer, and it can be upregulated or downregulated depending on its response to various exogenous or endogenous stimulators or suppressors. One shared feature of the investigated cancerous and noncancerous diseases is hypomethylation of the *HCP5* gene and upregulation of its transcript, which highlights the enormous potential of this lncRNA as a diagnostic biomarker in these pathologies. The *HCP5* gene sequence is a differentially methylated site associated with the epigenetic regulation of some disease phenotypes, such as obesity [64] and SLE [132], and it may also act in response to fetal development [128], aging [121], HIV [123] or influenza infection, and vaccination [124]. Presumably, hypomethylation of *HCP5* leads to added interactions and connectivity with protein and other RNA sequences, especially with microRNA regulators. Although increased *HCP5* levels seem to be a common event upon viral infection and in response to interferon stimulation and some cancers, the consequences of *HCP5* upregulation are diverse. The mode of *HCP5* action, whether acting as a promoter or suppressor, probably depends on the occurring downstream events. From the mechanistic point of view, *HCP5* either acts as a differentially methylated site involved in epigenetic regulation, or it is involved in transcriptional regulation of protein coding genes by sequestering miRNA as reported for a variety of different cancers (Table 6). Although transcriptional regulation by sequestration of transcription factors or regulatory miRNA seems to be the predominant mode of action by *HCP5* in cancerous diseases, it might also act as a specific or nonspecific ‘sponge’ for miRNAs in noncancerous diseases. The involvement of *HCP5* lncRNA in various human diseases and cancers underscores the importance of understanding its functions in the ceRNA networks as an important step towards future drug development. Perturbations in cellular regulatory functions due to interactions between *HCP5* and transcription factors probably contribute to some malignancies. For instance, recent studies of the interaction between *HCP5* and Runx family proteins suggests that they may play key roles in stem cell biology, particularly in regulating apoptosis and the G0/G1 transition by way of the ceRNA networks [150]. Runx1 increases the promoter activities and expression of *HCP5*, and it is also known to suppress HIV reactivation in T cells perhaps by way of *HCP5* mediation [166]. In many of the recent studies, the function of the *HCP5* gene is described as a defense-response gene often acting in unison with the inflammatory, innate, and adaptive immune response systems. However, *HCP5* also has disease progression and oncogenic effects that suggest that it may act like a double-edged sword to defend and to attack depending on other endogenous and exogenous regulators in the pathway.

The absence of the *HCP5* gene from the MHC genomic region of chimpanzees and gorillas and some human haplotypes that carry the *HLA-B*48:01* allele appears not to be deleterious or life-threatening [85]. Great apes and humans without the *HCP5* gene seem to live in relatively good health, which suggests that the deletion, if not harmful, might even confer some selective advantage [173]. Nevertheless, it is evident from an exploration of the recent literature and accumulating public databases that *HCP5* has many regulatory or associated functions at the genetic, molecular, and physiological levels. It appears to interact with numerous other genes and/or their products by way of intermediaries such as transcription factors and miRNA, *LTR16B2* genomic repeats, and as an enhancer or superenhancer in the regulation of other genes and in chromatid structural changes. There is still much to learn about the ceRNA actions of *HCP5*, and many of the future surprises about this hybrid MHC class I endogenous retroviral gene undoubtedly will arise from knockdown, knockout, and knockin gene expression studies. The human homologous *HLA-B*48:01* haplotype is a naturally occurring *HCP5-MICB* deletion or knockout haplotype in the human population that warrants genetic and epidemiological analysis to help elucidate the importance of these genes for most humans who still have the intact functional versions. Although the large databanks, datasets, and available publications have provided important insights into the functions of *HCP5*, much work still remains in order to elucidate the actual mechanisms and role of this intriguing MHC class I hybrid retroelement in immunity, health, and disease. More detailed studies are needed, particularly applying knockdown, knockout, and knockup studies, to find out more about its function as a regulator of various molecular and biological processes related to health and disease and the MHC.

## 9. Conclusions

*HCP5* is a unique human-specific gene within the MHC class I genomic region that encodes a hybrid HLA class I endogenous retroviral lncRNA with peptide coding potential. It has functional relationships with many other genes within or outside the MHC genomic region that are involved with antigen processing and presentation, the interferon regulatory pathway, and epigenomic and ceRNA networks; however, many of these functional interactions between multiple genetic variants are still poorly understood. *HCP5* gene SNVs and neighboring upstream and downstream SNVs have been associated with HIV viral load, HPV infection, autoimmune diseases, disease relapse after transplantation, and various cancers. Much still needs to be determined about the defensive and pathological functions of *HCP5* and its structural and functional role in RNA editing and signaling to enable epigenetic plasticity and immune response pathways. Judging from the recent new findings about the possible oncogenic role of *HCP5* as a ‘sponge’ for sequestering regulatory miRNA in cancer, new insights about its diverse mechanisms and functions in health and disease undoubtedly will continue to emerge and surprise in the near future.

## Figures and Tables

**Figure 1 cells-08-00480-f001:**
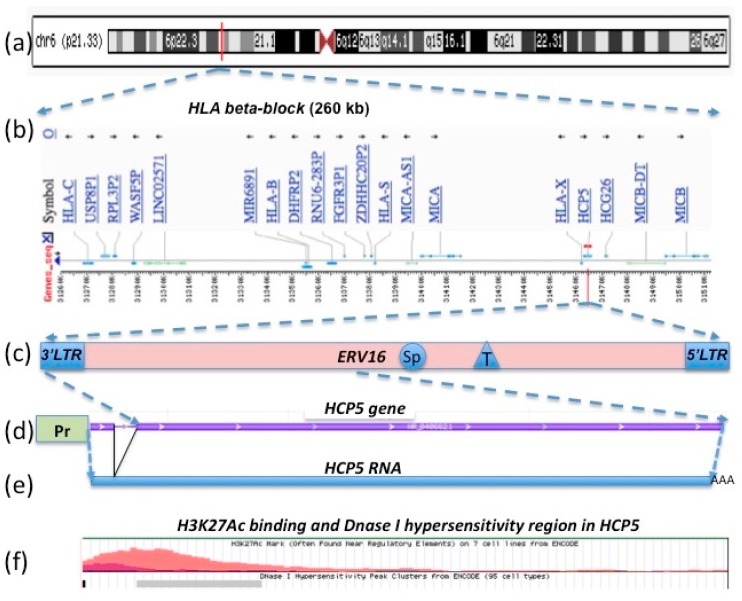
Location of the *HCP5* gene (**d**) within the *ERV16* element (**c**) and the human leukocyte antigen (HLA) class I region of the beta-block (**b**) on chromosome 6 at 6p21.33 (**a**). The HLA class I promoter region (the green rectangle labeled as Pr) for the 2630 bp *HCP5* gene (**d**) initiates transcription of the 2547 bp lnc *HCP5* RNA (**e**). The 91 bp intron in the *HCP5* gene is represented by the thin grey line between the violet rectangular lines (**d**). The *AluSp*, T*HE1B*, and *LTR162B* insertions within the 6173 bp *ERV16* sequence [30] (**c**) are indicated by the labeled circle, triangle, and rectangles, respectively (see Table 1 for more details). The H3K27Ac binding (orange curve) and Dnase I hypersensitivity region (grey horizontal rectangle) associated with the *HCP5* gene sequence (**d**) and sourced from the University of California, Santa Cruz (UCSC) genomic browser (Appendix A) are shown on line (**f**).

**Figure 2 cells-08-00480-f002:**
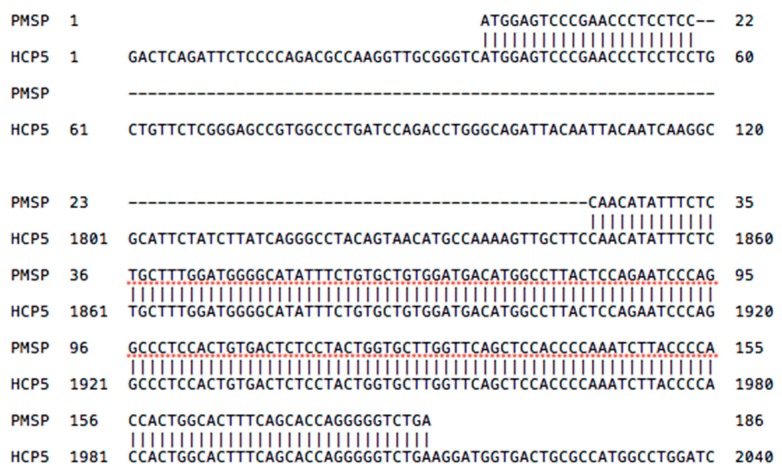
DNA nucleotide alignment between the papillomavirus minor structural protein (PMSP) (AJ437509.2) and *HCP5* RNA sequence (NR_040662.1).

**Figure 3 cells-08-00480-f003:**
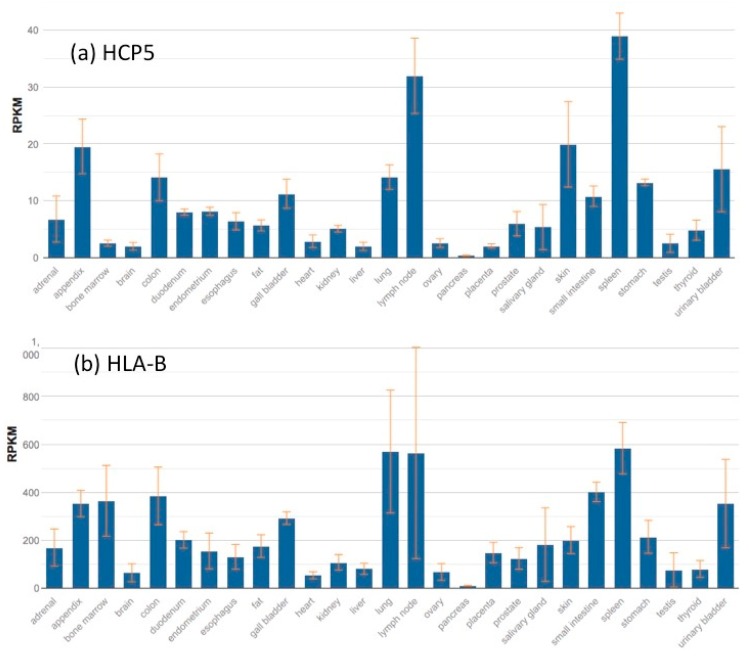
*HCP5* (**a**) and *HLA-B* (**b**) RNA sequences in 27 different normal tissues from 95 human individuals. NCBI BioProject PREJEB4337. RPKM (reads per kilobase of transcript per million mapped reads) on the *y*-axis is a normalized unit of transcript expression [60].

**Figure 4 cells-08-00480-f004:**
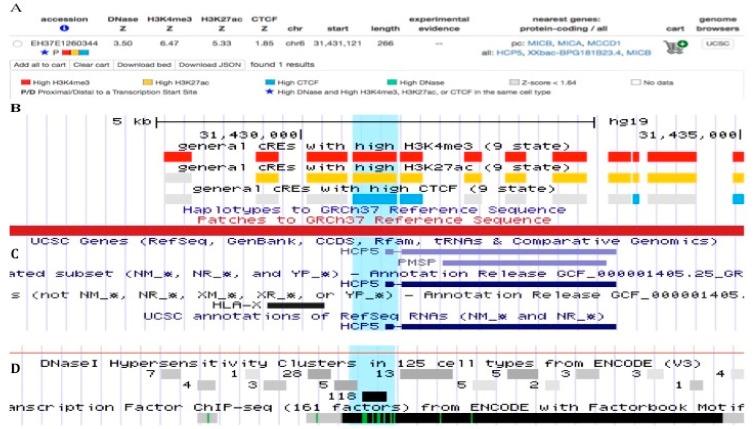
CTCF-binding sites proximal to and within the *HCP5* gene. (**A**) ENCODE accession number (EH37E1260344) and statistical scores for presence of binding sites for CTCF, H3K4me3, and H3K27ac and DNaseI hypersensitivity clusters. (**B**) The ENCODE regulation tracking details from the UCSC genome browser (Appendix A) show the CTCF locations (blue horizontal blocks on line 3) relative to (**C**) the *HCP5* and *HLA-X* gene positions below the horizontal red line. In (**B**) the cREs with a high probability for binding H3K4me3 are the red blocks on line 1, and those with high probability for binding H3K27ac are the yellow blocks on line 2. (**D**) The location of the DNaseI hypersensitivity clusters and 161 transcription factor binding sites from ChIP-seq data analysis archived in ENCODE are shown below the *HLA-X* and *HCP5* genes.

**Figure 5 cells-08-00480-f005:**
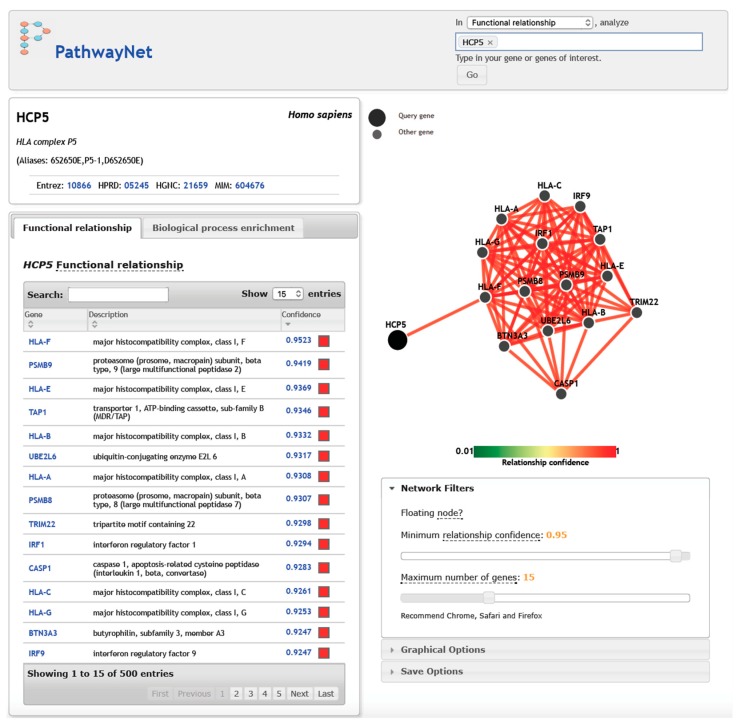
*HCP5* functional relationships in PathwayNet (Appendix A) [137].

**Table 1 cells-08-00480-t001:** The chromosomal location of *HCP5* and the *ERV16* (*LTR16B and ERV3-16A3*) retroelement between the pseudogene *HLA-X* and the *HCG26* lncRNA gene within the HLA class I region [30].

Sequence Name	Location on Chr6 *	Length bp	Orient.	Feature
MERC21	31,461,669–31,462,057	389	+	LTR fragment
HLA-X	31,461,846–31,462,490	645	+	Silent pseudogene
MERC21	31,462,544–31,462,625	82	+	LTR fragment
HLA-UTR	31,462,625–31,463,419	794	+	HLA-5′UTR + exon 1
HCP5-5′ncr	31,462,464–31,463,413	950	+	HLA class I promoter homologs
DNase region	31,463,001–31,463,190	190	+	100% reliability score
HCP5	31,463,180–31,465,809	2630	+	lncRNA gene
3′LTR16B	31,463,420–31,463,715	296	−	3′LTR (nt 150–446 in HCP5 RNA)
ERV3-16A3	31,463,716–31,464,933	1218	−	Internal (nt 447–2547 in HCP5 RNA)
AluSp	31,465,920–31,466,225	306	−	Insertion within ERV3-16A3
ERV3-16A3	31,466,237–31,467,707	1471	−	Internal
THE1B	31,467,708–31,468,050	343	+	Fragmented insertion within ERV3-16A3
ERV3-16A3	31,468,051–31,468,201	151	−	Internal
ERV3-16A3	31,468,362–31,468,644	283	−	Internal
5′LTR16B2	31,468,881–31,469,043	163	−	5′LTR
AluSx	31,469,044–31,469,282	239	−	Insertion within LTR16B
5′LTR16B2	31,469,283–31,469,592	310	−	5′LTR
L2	31,469,593–31,470,361	769	−	L2 LINE
MLT1E2	31,470,734–31,471,317	584	−	ERL-MaLR, LTR fragment
L2	31,471,325–31,472,047	723	−	L2 LINE
HCG26	31,471,229–31,472,408	1180	+	LncRNA

* Chr6:31461669-31472408, GRCh38.p12 assembly (annotation release 109 in March 2018). 5′ncr is 5′ noncoding region. Orientation of the sequence is on the +ve DNA strand or the −ve (complementary) DNA strand.

**Table 2 cells-08-00480-t002:** Effect of interferons and cytokines on HCP5 RNA expression (nd, no difference).

Cytokine	Cell Type	No: Controls/Tests	Up or Down	Geoprofile (206082_at)	Reference PMID
IL10	Peripheral blood mononuclear cells	4/4	down	GDS4551	23449998
KSHV-IRF4	lymphoma	2/2	up	GDS4956	24335298
IFN-alpha	natural killer cells	1/3	up	GDS4163	20334827
IFN-gamma	keratinocytes	5/4	up	GDS1846	14760888
IFN-alpha	hepatocytes (24 h)	3/3	up	GDS4390	22248663
IL28B	hepatocytes (24 h)	3/3	up	GDS4390	22248663
IFN-II	lung epithelium (24 h	4/4	up	GDS2341	16800785
IFN-gamma	bronchial epithelium (24 h)	5/5	up	GDS1256	15985639
IFNg + DXM	bronchial epithelium (24 h)	5/5	up	GDS1256	15985639
IFN-gamma	macrophages	6/6	up	GDS4232	22140520
IFN-gamma	microglia (24 h)	4/4	up	GDS1036	16163375
TNF	endothelial	4/4	nd	GDS1542	16617158
TNF	keratinocytes	4/4	up	GDS1289	15722350
IL1	macrophages	5/5	nd	GDS3005	18498781
IL6	macrophages	5/5	nd	GDS3005	18498781
IL6	macrophages	3/3	nd	GDS3290	18511485
Cigarette smoke	small airway cells	12/12	down	GDS2486	19106307

DXM is dexamethasone.

**Table 3 cells-08-00480-t003:** Diseases and phenotypes associated with *HCP5* SNV and retroelement (RE).

SNV ID	Chr Position	RE or Pseudogene	HCP5 Position	Ref	Disease or Phenotype
rs3094228	6:31462150	HLA-X	-	[91]	Thyroid peroxidase antibody positivity
rs115846244	6:31462283	HLA-X	-	[43]	Tonsillectomy
rs4360170	6:31462582	MERC21	-	[43]	Cold sores
rs3094605	6:31462917	HLA promoter	5’UTR	[92]	Lung cancer
rs3099840	6:31462944	HLA promoter	5’UTR	[92]	Lung cancer
rs3132090	6:31462975	HLA promoter	5’UTR	[92]	Lung cancer
rs2255221	6:31463914	ERV3-16A3	internal	[93]	HIV-1 control
rs2395029	6:31464003	ERV3-16A3	internal	[94]	Flucloxacillin drug liver injury
rs2395029	6:31464003	ERV3-16A3	internal	[33,34]	HIV-1 control
rs2395029	6:31464003	ERV3-16A3	internal	[95]	AIDS and HIV-1 control
rs2395029	6:31464003	ERV3-16A3	internal	[96]	Psoriasis and psoriatic arthritis
rs2395029	6:31464003	ERV3-16A3	internal	[93]	HIV-1 control
rs2395029	6:31464003	ERV3-16A3	internal	[97]	Abacavir-induced hypersensitivity
rs3130907	6:31464036	ERV3-16A3	internal	[92]	Lung cancer
rs2284178	6:31464348	ERV3-16A3	internal	[98]	MICB measurement
rs3094014	6:31465781	ERV3-16A3	3′UTR	[92]	Lung cancer
rs75640364	6:31465789	ERV3-16A3	3′UTR	[99]	Herpes zoster
rs3128986	6:31465916	ERV3-16A3	3′UTR	[92]	Lung cancer
rs79022003	6:31465935	AluSp	3’UTR	[100]	Eosinophil counts
rs3131620	6:31466054	AluSp	3′UTR	[92]	Lung cancer
rs3094604	6:31466334	ERV3-16A3	3’UTR	[101]	Squamous cell lung carcinoma
rs3094604	6:31466334	ERV3-16A3	3′UTR	[92]	Lung cancer
rs3094604	6:31466334	ERV3-16A3	3’UTR	[102]	MMR vaccine-related febrile seizures
rs3131619	6:31466554	ERV3-16A3	3’UTR	[103]	Myositis
rs3094013	6:31466589	ERV3-16A3	3’UTR	[104]	Idiopathic inflammatory myopathies
rs3131618	6:31466589	ERV3-16A3	3’UTR	[103]	Myositis
rs3094013	6:31466844	ERV3-16A3	3’UTR	[43]	Tonsillectomy
rs2523675	6:31468255	ERV3-16A3	3’UTR	[105]	Relapse after cord blood transplantation
rs2518028	6:31468270	ERV3-16A3	3’UTR	[105]	Relapse after cord blood transplantation
rs2523673	6:31469218	LTR16B/AluSx	3’UTR	[100]	Platelet count

**Table 4 cells-08-00480-t004:** Differentially methylated positions of HCP5′ in six recent studies.

Disease/Phenotype	CpG Site	Position Genecode	Feature	Delta	*p*-Value	Reference
Age-related expression	cg25843003	31431312	LTR16B2	−0.23	2.3 × 10^−13^	[121]
in monocytes and T cells	cg01082299	31431969	ERV3-16	−0.001	6.86 × 10^−6^	
SLE: Anti-dsDNA’	cg25843003	31431312	LTR16B2	−0.044	5.2 × 10^−10^	[122]
	cg00218406	31431407	LTR16B2	−0.067	8.7 × 10^−9^	
	cg01082299	31431969	ERV3-16	−0.04	1.2 × 10^−7^	
	cg18808777	31431503	ERV3-16	−0.051	3.0 × 10^−10^	
HIV associated marker	cg00218406	31431407	LTR16B2			[123]
Influenza vaccination	CpGsite	31428956	MER21B	−0.301	1.31 × 10^−4^	[124]
		31431456	LTR16B2			
		31431457	LTR16B2	−0.401	2.0 × 10^−7^	
		31433586	3′UTR			
Endometrium	cg25843003	31431312	LTR16B2	−0.064		[125]
Pre to receptive	cg00218406	31431407	LTR16B2	−0.058		
	cg21684411	31431573	ERV3-16	−0.071		
BMI	cg00218406	31431407	LTR16B2	0.005	8.03 × 10^−7^	[64]
BMI	cg25843003	31431312	LTR16B2	0.002	1.51 × 10^−6^	
Obesity	cg00218406	31431407	LTR16B2	0.048	4.34 × 10^−5^	
Obesity	cg25843003	31431312	LTR16B2	0.020	5.98 × 10^−5^	

The negative delta values reflect hypomethylation and gene upregulation and the positive delta values reflect hypermethylation and gene downregulation. SLE is Systematic Lupus Erythematosus. The p-values were mostly False Discovery Rate (FDR) adjusted.

**Table 5 cells-08-00480-t005:** Top 52 *HCP5* gene interactions associated with Comparative Toxicogenomics Database (CTD) studies.

Gene Symbol	Gene ID	Interaction Count	Gene Symbol	Gene ID	Interaction Count
PTGS2	5743	71	ICAM1	3383	9
TNF	7124	55	NOS3	4846	9
IL1B	3553	52	PON1	5444	9
PTGS1	5742	42	SCARB1	949	9
CASP3	836	28	AHR	196	8
BCL2	596	21	APOA1	335	8
AGT	183	20	TXN1	22166	8
CTNNB1	1499	20	VEGFA	7422	8
CCND1	595	18	ABCA1	19	7
NFKB1	4790	18	ALB	213	7
RELA	5970	17	CCL2	6347	7
TNFSF10	8743	17	IKBKB	3551	7
NOS2	4843	14	ITGA2B	3674	7
PARP1	142	14	MAPK1	5594	7
NFKBIA	4792	13	MAPK3	5595	7
BAX	581	12	MYC	4609	7
CDKN1A	1026	12	PPARG	5468	7
IL4	3565	12	SELP	6403	7
IL6	3569	11	TCF4	6925	7
ITGB3	3690	11	ABCB1	5243	6
MMP9	4318	11	AKT1	207	6
CASP9	842	10	CYSLTR1	10800	6
IL1A	3552	10	JUN	3725	6
PPARA	5465	10	MMP2	4313	6
BIRC5	332	9	PLAUR	5329	6
CASP8	841	9	RB1	5925	6

**Table 6 cells-08-00480-t006:** *HCP5*–miRNA–protein coding gene regulator interactions.

Antioncogenic miRNA	HCP5 Action	Gene Symbol and Up- or Downregulated	Cancer Type	Reference
miR-203	sponge	TGFb/SMAD3	Lung	[56]
miR-106b-5p	network	CTSS/FGL2	Lung	[130]
miR-17-5b	network	PDCD1LG2/PDL2	Lung	[130]
miR-126		GSR/ASCL1 ↑	Lung metastasis	[152]
		MET/GRM8/DACH1 ↓	Lung metastasis	[152]
miR-139	absorption	RUNX1 ↑	Glioma	[150]
miR-139-5p		ZEB1	Colorectal	[153]
		APIG1 ↓	Colon	[154]
miR-22-3p	sponge	ST6GAL2 ↑	Thyroid	[155]
mi-186-5p	sponge	ST6GAL2 ↑	Thyroid	[155]
miR-216a-5p	sponge	ST6GAL2 ↑	Thyroid	[155]
miR-15a	adsorption	MACC1 ↑	Cervical	[156]
miR-155	sponge	Complement genes	Breast	[157]
miR-128	sponge	MACC1 ↑	Gastric	[163]
miR-101	sponge	NDUFB6 ↓	Gastric	[163]
miR-103a	sponge	NDUFB6 ↓	Gastric	[163]
24 different mIR	ceRNA	TAP1 ↑	Lymphoma	[164]
	ceRNA	PSMB9 ↑	Lymphoma	[164]
	ceRNA	KLF2 ↑	Lymphoma	[164]
	ceRNA	GIPC1 ↑	Lymphoma	[164]
	ceRNA	ETN3A3 ↑	Lymphoma	[164]
	ceRNA	ETN3A1 ↑	Lymphoma	[164]
	ceRNA	CD47 ↑	Lymphoma	[164]
	ceRNA	CCDC50 ↑	Lymphoma	[164]
	ceRNA	LMBR1L ↑	Lymphoma	[164]
	ceRNA	HERC6 ↑	Lymphoma	[164]

↑ is upregulation and ↓ is downregulation.

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
