# Peer review of "Long Noncoding RNA HCP5, a Hybrid HLA Class I Endogenous Retroviral Gene: Structure, Expression, and Disease Associations"

_cells, 2019, doi:10.3390/cells8050480_

Round 1

Reviewer 1 Report

 This manuscript consists of a review of the current state of knowledge on the retroviral-originating lncRNA antisense to HLA-B: HCP5. This is a well-crafted and comprehensive review of information pertaining to this lncRNA and will make an excellent addition to this special issue.

Minor points

Line 49, can define microRNA as miRNA at this point.

Line 65, I don’t understand the use of “centromeric of the two classical...”, does the author simply mean “ in between” the two genes?

Line 97, it would be nice to briefly describe the what the Harmonizome is.

Line 105, not sure how broad the expected audience is, but the terms alpha, beta, kappa and epsilon frozen blocks might not mean much to most readers.

Line 494, rs2395029 refers to a specific SNP, it’s a bit confusing to write “with a rs2395029 amino acid substitution”.

Line 560, there is an isolated “ without a partner.

Author Response

Reply to reviewer 1

Thank you for the review and suggested revisions.

Line 49, can define microRNA as miRNA at this point.

This has been corrected and marked on the revised version.

Line 65, I don’t understand the use of “centromeric of the two classical...”, does

the author simply mean “ in between” the two genes?

Changed to:  ‘downstream, at the centromeric end, of the two classical HLA class I genes HLA-B and HLA-C (Fig. 1) [29].’

Line 97, it would be nice to briefly describe what the Harmonizome is.

Added the following: ‘…Harmonizome that is an integrated knowledge-base connecting big data with a collection of information about genes and proteins from 114 datasets provided by 66 online resources (Table S1).’

Line 105, not sure how broad the expected audience is, but the terms alpha, beta, kappa and epsilon frozen blocks might not mean much to most readers.

Have deleted the terms alpha, beta, kappa and epsilon in the revised manuscript.

Line 494, rs2395029 refers to a specific SNP, it’s a bit confusing to write “with a

rs2395029 amino acid substitution”.

Changed to ‘…with an amino acid substitution at the SNP rs2395029[28,30].’

 Line 560, there is an isolated “ without a partner.

The quote between “ and “ has been italicized for added clarity:the use of HCP5 rs2395029 testing could be as useful as HLA-B*57:01 typing to prevent the abacavir hypersensitivity reaction. Given that HCP5 testing is cheaper, less time-consuming and easier to perform than HLA typing, it may confidently replace the latter in clinical settings.

Reviewer 2 Report

This is a very extensive and comprehensive review about the HCP5 lncRNA, covering the organization of its genomic locus, its gene regulation, the association with diseases, and the molecular mechanisms HCP5 is involved in. Clearly, the author has put a lot of effort in gathering all the information from the literature and public available datasets. I consider this review as a very useful compendium for researchers working in the field. If at all, my only concern would be the length of the manuscript, which might be confusing to some readers. The only solution I can envision would be to split the current manuscript into two reviews (one about the genomics and regulation, the other about the disease-association), however, this might not fit the author's initial intention.

Author Response

Reply to Reviewer 2

Thank you for the review and suggested revision.

If at all, my only concern would be the length of the manuscript, which might be confusing to some readers. The only solution I can envision would be to split the current manuscript into two reviews (one about the genomics and regulation, the other about the disease-association), however, this might not fit the author's initial intention.

As the author, I did worry about its length, but Cells editorial office assured me that there is no restriction on the length of submitted papers. Therefore, I prefer to leave it at its current length rather than splitting into two reviews as suggested by the reviewer.